# DPD-LoRA: Dynamic Prompt-Driven Low-Rank Adaptation for Improved Generalization

## Abstract

Fine-tuning large models presents technical challenges such as catastrophic forgetting and parameter inefficiency. Low-rank Adaptation (LoRA) and Propmt Learning can help address some of these challenges by providing more compact and flexible representations. However, Low-rank approximation is susceptible to outliers and relies on the assumption of a global low-rank structure, which can be suboptimal. Additionally, Prompt learning can overfit to specific downstream tasks, reducing its effectiveness when adapting to new tasks. In this paper, we introduce **Dynamic Prompt-Driven Low-Rank Adaptation (DPD-LoRA)**, a novel framework that seamlessly integrates task-specific guidance using hierarchical prompt tokens and parameter-efficient adaptation. Unlike traditional methods, task-aware prompts in the DPD-LoRA dynamically influences low-rank updates in the model's parameters, thus enabling robust adaptation and generalization across diverse tasks and mitigating the forgetting issues. We further improve the learning capabilities of the model by breaking down the standard LoRA into multiple low-rank sub-matrices, without adding additional parameters. Further, we use an adaptive loss function to guarantee alignment with the distribution of the pre-trained model. Specifically, we introduce a self-regulated mechanism to improve stability, and a soft-gated selection mechanism to decide when to activate adaptation modules to improve performance on unseen categories. Extensive experiments on 11 benchmark datasets demonstrate that DPD-LoRA significantly outperforms state-of-the-art methods in both accuracy and generalization, offering a comprehensive solution to the challenges of fine-tuning large-scale models.

## 1 Introduction

Large Models (LMs) have demonstrated remarkable capabilities across various domains, often achieving state-of-the-art performance. Their robust zero-shot and few-shot learning abilities have positioned them as foundational models in both Natural Language Processing (NLP) and Computer Vision (CV). However, their enormous parameter counts pose computational challenges for training and fine-tuning. For instance, BERT (Devlin et al., 2018) contains approximately 300 million parameters, GPT-3 (Brown et al., 2020) has 175 billion parameters, and the visual model SAM (Kirillov et al., 2023) possesses 632 million parameters. This trend extends to Vision-Language Models (VLMs) like LLaVA (Liu et al., 2024a), which range from 7 billion to 70 billion parameters, following the scaling laws in machine learning.

To address this issue, research topics differ in two mainstreams. The parameter-efficient fine-tuning (PEFT) was earlier discovered to fine-tune pre-trained models with only a few parameters. Among PEFT methods, LoRA(Hu et al., 2021) is an undoubtedly notable method that decomposes pre-trained weights into two low-rank matrices and trains only these low-rank matrices, without changing the structure of the original model. Consequently, LoRA can outperform full fine-tuning in some cases. However, LoRA faces limitations, including the risk of catastrophic forgetting of pre-trained knowledge and overfitting to specific downstream tasks (Kalajdzievski, 2024). To address these shortcomings, we explore the integration of task-specific guidance into the adaptation process. One promising direction is the use of prompts, which have been successful in guiding large models to perform new tasks without modifying their parameters significantly.

Building on this, prompt learning, initially developed in the NLP community, uses strategically designed prompts to steer models towards desired behaviors. This approach has been extended to CV

Figure 1: **Dynamic Prompt-Driven LoRA (DPD-LoRA)** demonstrates outstanding quantitative and qualitative performance. (1a): DPD-LoRA outperforms prior state-of-the-art techniques across various image recognition datasets, particularly excelling in novel class generalization. (1b): DPD-LoRA showcases accelerated convergence and favorable early-stage performance, surpassing previous state-of-the-art benchmarks within just 10 epochs.

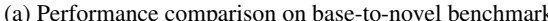

(a) Performance comparison on base-to-novel benchmark

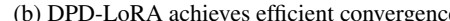

(b) DPD-LoRA achieves efficient convergence

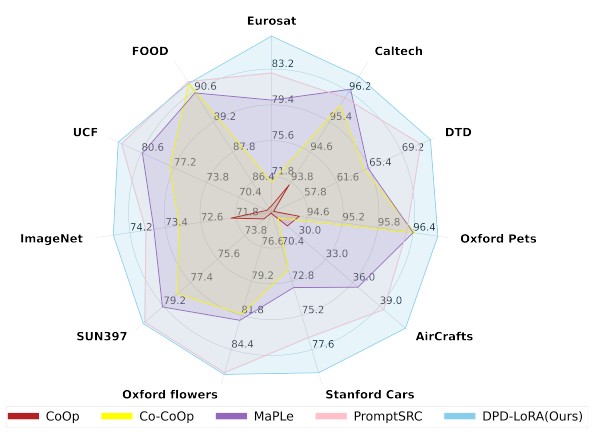

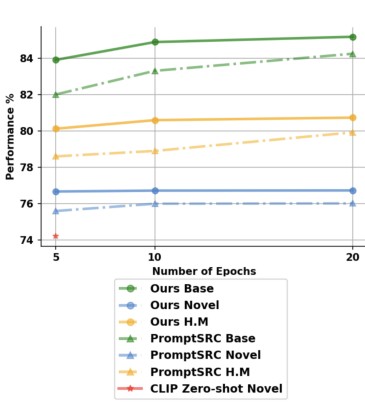

and multimodal tasks, where learnable prompts serve as adaptable input modifications that tailor pre-trained models to new tasks with minimal additional training. However, employing prompts alone also presents challenges. Deep prompt tokens are often randomly initialized and may lack coherence across layers, leading to suboptimal guidance(Xu et al., 2023). Additionally, prompts primarily influence the input representation, without directly modifying the model's internal computations such as attention mechanisms, leading easily over-fitting in training class (Zhou et al., 2022a; Khattak et al., 2023b). This means that while prompts can guide the model's focus, they may not fully address the adaptation and generalization challenges.

Recognizing the limitations of existing approaches, we propose a new paradigm for adapting large models that incorporates task-specific guidance directly into the adaptation mechanism. Our method introduces a dynamic adaptation process where task-aware signals influence the low-rank updates of the model's parameters. This not only enhances the model's ability to generalize across diverse tasks but also mitigates issues like catastrophic forgetting and overfitting. Unlike traditional prompt learning, which primarily modifies input representations, our approach embeds task-specific information within the model's internal computations. By extending adaptation modules into the lower-rank subspaces without increasing the number of parameters, we enable a more coherent and effective learning process. Our framework also includes a self-regulated mechanism to maintain subspaces consistency, an adaptive loss function for alignment with the pre-trained distribution, and a soft gated selection mechanism to optimize performance on unseen categories. Extensive base-to-novel experiments shows that our method outperforms the competitive baseline approaches, as shown in Fig. 1a. Also, the DPD-LoRA shows faster converge speed, as evidenced in Fig. 1b.

Our contributions are summarized as follows:

- We introduce a novel approach that leverages dynamic prompts-driven LoRA matrices. We demonstrate that LoRA layers can be oriented towards particular downstream tasks through Prompts Learning.

- We present a self-adaptive loss, based on a pretrained LoRA model, to regulate the distribution of Lower-rank SubSpaces (LoRSS), which mitigates overfitting in extended training.

- We propose a Prompt-Conditioned Gating Mechanism (PCGM) that assigns soft weights to each LoRA layer, dynamically adjusting LoRA's contributions based on the prompt.

- We evaluate our proposed pipeline on 11 benchmark datasets, **without any additional models prior**, achieving competitive quantitative and qualitative results.

## 2 RELATED WORK

**VLMs**:Recent developments in foundational vision-language models (VLMs) suggest that multi-modal learning has significantly advanced through the integration of paired images and text. Utilizing massive image-text pair datasets from the web, CLIP(Radford et al., 2021) employs contrastive learning to enable the model to understand similarities between the visual and textual branches. Similar methodologies are adopted in ALIGN(Jia et al., 2021), LiT(Zhai et al., 2022), FLIP(Li et al., 2023), and Florence(Yuan et al., 2021). These notable approaches involve scaling up the models by using large amounts of data, increasing batch sizes, and enhancing model dimensions. Consequently, these models provide robust few-shot or zero-shot capabilities due to their enormous parameter sizes. However, a large number of parameters also introduces a side-effect; it becomes impractical to adapt these models to challenging downstream tasks. Our study is motivated to integrate Parameter-Efficient Fine-Tuning (PEFT) methods and Prompt Learning to minimize the need for parameter tuning in pretrained models while maintaining their generalization capabilities.

**PEFT**: PEFT reduces the cost of fine-tuning large models by updating a small subset of parameters. Beyond prompt learning, PEFT methods mainly involve two approaches: adapter-based learning and low-rank adaptations. Adapter-based learning inserts modules into a frozen backbone (Houlsby et al., 2019; Mahabadi et al., 2021; He et al., 2021) but can increase inference latency. Low-rank adaptations like LoRA (Hu et al., 2021) and its variants (Liu et al., 2024b) introduce low-rank matrices to approximate weight updates, integrating them with frozen weights during inference. Zhang et al. (2023) enhances this by using SVD decomposition and pruning insignificant singular values. Subsequent variants like QLoRA (Dettmers et al., 2024) combine quantization with low-rank adapters, and VeRA (Kopiczko et al., 2023) uses vector-based random matrices for adaptation. However, dependency on model dimensions limits scalability. To address this, LoRA-XS (Bałazy et al., 2024) employs SVD of pre-trained weights to initialize and freeze projection matrices, greatly reducing trainable parameters independently of model size. Similarly, SVFT (Lingam et al., 2024) leverages outer products of singular vectors, training only coefficients to achieve higher accuracy and fine-grained control. Both methods emphasize the role of singular vectors in weight updates, suggesting potential for further SVD-based PEFT advancements.

**Prompt learning**: Prompts initially started with discrete, human-crafted templates in the NLP community, such as "a photo of a <CLASS>" for CLIP-like models(Cherti et al., 2023). However, designing an appropriate prompt can be abstract, as it is challenging to determine which types of prompts best fit specific tasks. Recent methods propose treating prompts as learnable vectors in an end-to-end manner while keeping the model's parameters frozen. Specifically, CoOP(Zhou et al., 2022b) demonstrates that randomly initializing a set of vectors as prompts in the text encoder of the CLIP model achieves results comparable to those of human-designed prompts. Furthermore, CocoOp(Zhou et al., 2022a) and MaPLE(Khattak et al., 2023a) further utilize prompts on the image encoder. The former employs a lightweight network on the image encoder to generate an instance-wise constraint for textual-branch prompts, while the latter adopts similar ideas from VPT(Jia et al., 2022) to innovatively add prompts on both visual and linguistic branches with a coupled relationship. Although all these methods demonstrate the effective transfer of prior knowledge from CLIP, they may also negatively affect generalization capabilities. Thus, PromptSRC(Khattak et al., 2023b) proposes a self-regulated strategy to minimize the divergence between the zero-shot CLIP and Prompted CLIP models. And many other approaches(Li et al., 2024) utilize more powerful models to provide prior to increase model's capability, or like(Yao et al., 2023; Roy & Etemad, 2023) introduce Large language models to implement augmentations in textual branch to help generalization.

## 3 METHOD

In this section, after reviewing the classic vision-language model CLIP in Section 3.1, we introduce the key components of our proposed method, DPD-LoRA. In Section 3.2, we explain how Prompts and LoRA influence self-attention and guide LoRA within the Multi-Head Attention mechanism. Following this, in Section 3.3, we adapt Hierarchical interaction for prompt tokens and LoRAs across multiple modalities. Additionally, we introduce our novel Prompt-Conditioned Gate in Section 3.4, which strengthens the connection between prompt and LoRA layers. Finally, we introduce a novel self-constraining method in Section 3.5, which helps prevent overfitting of the proposed DPD-LoRA on downstream tasks and enhances the model's zero-shot capabilities. The combina-

Figure 2: **Dynamic Prompt-Driven Low-Rank Adaptation (DPD-LoRA) Framework:** DPD-LoRA framework uses prompts to guide low-rank adaptation. Initially, LoRA is learned after each multi-head attention block, which decomposed feature space into the multiple lower-rank subspaces(LoRSS), as illustrated in Fig. 3. Additionally, self-regularized component is introduced, that apply an early-stopped LoRA (extended to LoRSS as well) and fix it as an anchor to constrain the current model's distribution, as detailed in Sec. 3.5. Moreover, a prompt-conditioned gating mechanism (PCGM) is introduced to strengthen connections between prompts and LoRSS. Finally, DPD-LoRA promotes interactions among prompt tokens, which is accomplished through a hierarchical interaction approach.

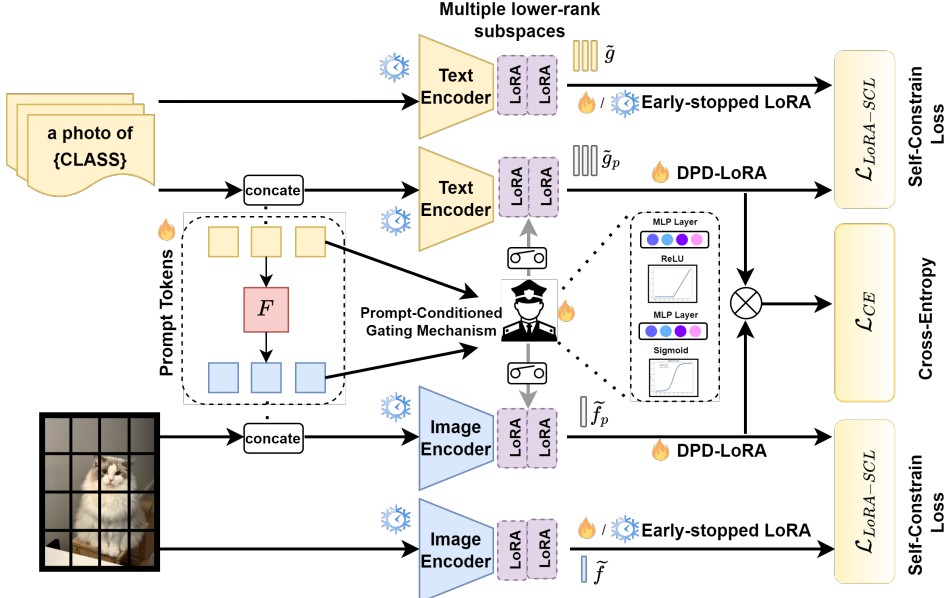

tion of these components forms a cohesive framework that addresses both learning efficiency and robustness. Our overall framework is illustrated in Fig. 2.

## 3.1 PRELIMINARIES

We adopt the pretrained CLIP as our foundational vision-language (VL) model in our methods. These types of VL models typically consist of two parallel encoders: a transformer(Dosovitskiy et al., 2020a) for text encoding and either a ResNet(He et al., 2016) or ViT(Dosovitskiy et al., 2020b) as the visual encoder. Suppose we denote the pretrained parameters $\{\theta_v, \theta_t\}$ for the visual and text encoders, respectively. Specifically, on the **visual** side, the input image $X$ is divided into patches followed by a projection to produce patch tokens. Furthermore, an additional learnable class token $e_{cls}$ is introduced to the patches list, rewriting the input list as $\widetilde{X} = \{e_1, e_2, \ldots, e_n, e_{cls}\}$. The visual encoder $f$ then generates the image feature $\widetilde{f} = f(\widetilde{X}, \theta_v)$ according to the input list through multiple transformer blocks. On the **textual** branch, human-designed templates contain corresponding labels, such as 'a photo of a {class}', to represent the raw input $Y$. Following this, CLIP tokenizes the words and projects them into word embeddings to create a textual input list $\widetilde{Y} = \{t_{sos}, t_1, t_2, \ldots, t_L, t_{cls}, t_n, t_{eos}\}$, where $t_{sos}$ and $t_{eos}$ refer to the learnable start and end tokens of the sentence, respectively, and $t_L$ represents the embedded template. Similarly to the visual branch, the text encoder $g$ generates the text feature $\widetilde{g} = g(\widetilde{Y}, \theta_t)$ through multiple transformer blocks. Based on contrastive learning, for class labels ranging from 1 to C, we can formulate the zero-shot inference equation as follows:

$$p(\hat{y}|X) = \frac{\exp((\widetilde{g} \cdot \widetilde{f})/\tau)}{\sum_{i=1}^{C} \exp((\widetilde{g_i} \cdot \widetilde{f})/\tau)}, \tag{1}$$

where $\tau$ is the temperature parameter, and $\hat{y}$ is the predicted label corresponding to the input image $X$ that exhibits the highest cosine similarity score, denoted as $\mathrm{sim}(\cdot)$.

Figure 3: **Expanded Lower-Rank Sub Space (LoRSS):** The proposed approach involves the gradual integration of LoRSS. DPD-LoRA extends the LoRA concept to the multiple lower-rank feature spaces within multi-head attention. Each LoRSS is forced to interact with its preceding LoRSS, if applicable, as described in Eq. 4. Note that only the parameters of LoRSS, Prompts, and PCGM are trained, while all other parameters remain frozen.

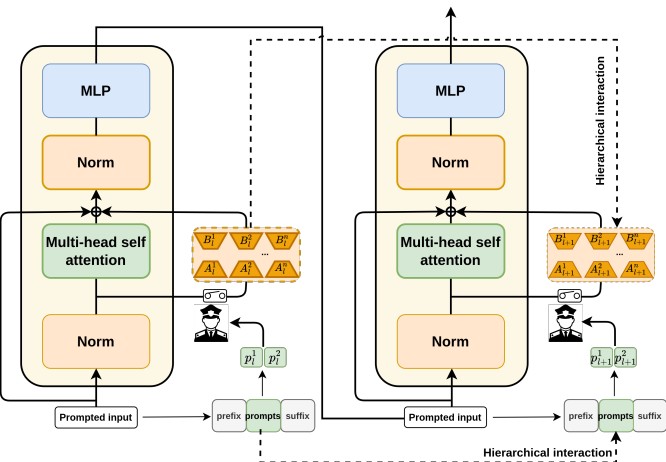

### 3.2 PROMPT LEARNING WITH LOW RANK ADAPTATION IN TRANSFORMERS

In this paper, we adopt the techniques described in both (Khattak et al., 2023b;a) as our baseline model. Prompt Learning involves introducing a set of learnable tokens, which are appended to the original input. This can be implemented in the text branch with Textual Prompts $P_t = \{p_t^1, p_t^2, \ldots, p_t^n\}$ (Zhou et al., 2022b), or in the visual branch with Visual Prompts $P_v = \{p_v^1, p_v^2, \ldots, p_v^n\}$ (Jia et al., 2022), or simultaneously in both branches (Khattak et al., 2023a). The new inputs, incorporating these prompts, replace the original text and visual inputs, yielding $\widetilde{Y}_p = \{t_{sos}, P_t, t_1, t_2, \ldots, t_L, t_{cls}, t_n, t_{eos}\}$ for text and $\widetilde{X}_p = \{P_v, e_1, e_2, \ldots, e_n, e_{cls}\}$ for visuals. Their interaction with the Multi-Head Attention (MHA) mechanism in transformers is as follows($\phi$:softmax function and $X'$: prompted inputs):

$$\text{MHA}(X') = \text{Concat}(\text{head}_1, \ldots, \text{head}_n)W_o, \quad \text{head}_i = \text{Attn}(Q_i, K_i, V_i) = \phi\left(\frac{Q_i K_i^T}{\sqrt{d_h}}\right) V_i \quad (2)$$

Among this sitting, we then integrate LoRA by adjusting the weight matrices with low-rank updates, where $A \in \mathbb{R}^{d_{model} \times r}$ and $B \in \mathbb{R}^{r \times d_{model}}$ are learnable matrices with rank $r \ll d_{model}$. Applying LoRA to the attention mechanism, the modified output becomes:

$$h = X'W + X'\Delta W = X'W^{base} + X'AB. \quad (3)$$

Here, $h$ is the new output generated by the prompted input $X$ and the LoRA-adjusted weights. This integration allows the model benefits from task-specific prompts that guide attention mechanisms to focus on relevant features, while LoRA provides efficient fine-tuning of the model's weights. This synergy enhances performance without significant computational overhead. For a detailed mathematical derivation and further explanations, please refer to appendix D.

### 3.3 EXPANDED SUBSPACES AND HIERARCHICAL INTERACTION

Although plain LoRA itself has shown promising results, an expanded lower-rank feature subspace appears to strengthen the model under the same parameter settings. We decompose a single LoRA into multiple sub-LoRAs, forming a Lower-Rank SubSpace (LoRSS). This transformation modifies the original LoRA (Eq. 3) into LoRSS (Eq. 4), where $m$ refers to the number of sub-LoRAs, and $s_i$ refers to scaling factors corresponding to each sub-LoRA.

$$h = WX' + X'\Delta W = WX' + X'\sum_{i=1}^{m}(s_i * A_i B_i) \quad (4)$$

Existing research on prompt learning, which utilizes deep prompt tokens, has typically involved generating these tokens independently from a Gaussian distribution. Each deep token does not communicate with others, as referenced in various studies (Jia et al., 2022; Khattak et al., 2023a;b). However, these methods naturally ignore dependencies within the information. To prevent information loss, DPD-LoRA enforces that each token interacts with its previous token to guide our LoRSS more effectively. Now, the prompt token becomes a hierarchically coupled prompt as defined in Eq. 5, where $\alpha$ is a hyperparameter that weights different layer tokens.

$$X^{'} = [X, P] = [X, \alpha * P^l + (1 - \alpha) * P^{l-1}] \tag{5}$$

Here, $[X, P]$ denotes the concatenation of the input $X$ and the combined prompt $P$. The term $l$ and $l-1$ represents current layer and previous layer, introducing hierarchical interaction between tokens. Building on this concept, DPD-LoRA further extends hierarchical interaction to the LoRA layers in the multi-modal domain. Our method not only utilize the tokens themselves but also modify each LoRA layer to interact with its preceding layer, thereby preventing information loss across layers. This process is formalized in Eq. 6, where $\alpha$ is same hyperparameter that assign weights to different layers. Thus, the LoRA layers can access information from the previous layer.

$$\Delta W_l = \alpha \sum_{i=1}^{m} \left(s_i^{(l)} * A_i^{(l)} B_i^{(l)}\right) + (1 - \alpha) \sum_{i=1}^{m} \left(s_i^{(l-1)} * A_i^{(l-1)} B_i^{(l-1)}\right) \tag{6}$$

In this equation, the first term represents the contribution from the current layer $l$, while the second term incorporates information from the previous layer $l - 1$. By adjusting $\alpha$, model can balance the influence of the current and preceding layers.

### 3.4 PROMPT-CONDITIONED GATING MECHANISM (PCGM)

Effective adaptation in large models requires a cohesive integration of task-specific information into the model's internal parameters. A straightforward approach might involve independently applying prompts to guide LoRA layers without coordination. However, such methods lack synergy between the input prompts and the adaptation layers, potentially leading to suboptimal performance, as the ablation study shows in Tab. 5.

To address this, we propose a soft-gated mechanism, **Prompt-Conditioned Gating (PCGM)**, that dynamically modulates LoRA layers based on task-specific prompts. Specifically, DPD-LoRA introduces a gating function $G(\cdot)$, which takes prompt tokens $P$ as input and outputs gating values for each LoRA layer, controlling their contributions during inference. Acting as a bridge between prompt embeddings and LoRA adaptations, the gating function ensures task-relevant information directly influences internal computations. Implemented as a learnable mapping, this mechanism promotes interaction between prompts and parameter updates through a shared gate, fostering coherent and synergistic adaptation. Using a single gating function across prompts ensures consistent control, improving generalization while maintaining efficiency with minimal computational overhead and no increase in parameter count. The updated equation becomes:

$$\Delta W_l = \Delta W_l * G(P_l); P_l \in P \tag{7}$$

### 3.5 SELF-REGULARIZED LOWER-RANK SUBSPACES(LORSS)

Since both LoRA and Prompt Learning are task-specific PEFT techniques, they are in our case optimized solely with the cross-entropy loss function $\mathcal{L}_{CE}$. As a result, they can easily lead to overfitting, which causes the pre-trained models to lose their generalization ability. Particularly, as the number of training epochs increases, these techniques tend to diverge from the pre-trained knowledge and overly specialize on supervised downstream tasks. This often results in the fine-tuned model scoring well in baseline evaluations but performing poorly in novel evaluations.

Inspired by PromptSRC (Khattak et al., 2023b), which introduced an anchor prompt training method that self-constrains the fine-tuned model using its fully frozen counterpart, we propose a self-regularization term to prevent drift in our LoRA distribution. Specifically, DPD-LoRA duplicates each encoder in both branches and retain the unprompted inputs, i.e., the unperturbed embeddings $\widetilde{Y}$ and $\widetilde{X}$, to train the LoRA-CLIP model exclusively. After several epochs, we implement early stopping to prevent overfitting and stabilize the Lower-Rank SubSpaces (LoRSS) distribution ($D_{\text{LoRSS}}$).

To further prevent overfitting and encourage diversity in the learned LoRA feature space, we adopt an orthogonal regularization loss term. The regularization loss $\mathcal{L}_{\text{orth}}$ is mathematically defined in

Eq. 8, where the $N$ is the number of LoRA parameters being regularized. $\delta$ represents a LoRA parameter matrix.

$$\mathcal{L}_{\text{orth}} = \frac{1}{|N|} \sum_{\delta \in N} \begin{cases} \left\| \delta\delta^\top - I \right\|_F, & \text{if } \delta \text{ corresponds to } \texttt{A} \\ \left\| \delta^\top\delta - I \right\|_F, & \text{if } \delta \text{ corresponds to } \texttt{B} \end{cases} \tag{8}$$

Finally, to ensure that the DPD-LoRA model adheres to the stabilized LoRSS distribution ($D_{\text{DPD-LoRA}}$), we employ a Kullback-Leibler divergence loss, as shown in Eq. 9. This self-regularization approach effectively mitigates overfitting issues, leading to improved performance as illustrated in Fig. 1b. Here, $\lambda_1$ and $\lambda_2$ refer to regularization term weights.

$$\mathcal{L}_{\text{SCL-LoRA}} = \lambda_1 \mathcal{D}_{kl}(D_{\text{LoRSS}} || D_{\text{DPD-LoRA}}) + \lambda_2 \mathcal{L}_{\text{orth}} \tag{9}$$

In addition, to further enhance the expressiveness of the model, we incorporate a text diversity loss ($\mathcal{L}_{\text{text-diversity}}$), which was initially proposed in PromptSRC to promote greater diversity in the textual branch. Thus, our final training objective becomes:

$$\mathcal{L}_{\text{total}} = \mathcal{L}_{\text{CE}} + \mathcal{L}_{\text{SCL-LoRA}} + \lambda_3 \mathcal{L}_{\text{text-diversity}} \tag{10}$$

where $\mathcal{L}_{\text{CE}}$ represents the cross-entropy loss, $\mathcal{L}_{\text{SCL-LoRA}}$ is the self-regularization component introduced earlier, and $\lambda_3$ is a weight balancing the impact of the text diversity loss. This comprehensive objective aims to facilitate adaptation to downstream tasks while constraining overfitting.

## 4 EXPERIMENTS

We conducted extensive experiments on 11 datasets to assess the effectiveness of our proposed method. The experiments demonstrate that incorporating prompt learning and the lower-rank subspaces improves model performance. Furthermore, a complementary relationship can be observed between prompt vectors and the lower-rank subspaces. Our results show favorable performance on most datasets and achieve state-of-the-art performance on average.

### 4.1 DATASETS:

In this study, we apply our methods to 11 datasets, following the approaches previously described in (Zhou et al., 2022b;a; Khattak et al., 2023a). These applications facilitate base-to-novel and cross-dataset experiments for the image recognition task. We include general object recognition datasets such as ImageNet (Deng et al., 2009) and Caltech (Fei-Fei et al., 2004). For a more detailed evaluation that focuses on fine-grained image features, we utilize datasets such as FGVC-Aircraft (Maji et al., 2013), Stanford Cars (Krause et al., 2013), Flowers102 (Nilsback & Zisserman, 2008), Food101 (Bossard et al., 2014), and Oxford Pets (Parkhi et al., 2012). Additionally, we assess the robustness and adaptability of our methods across various environmental and contextual conditions using datasets for scene and action recognition. These include SUN397(Xiao et al., 2010) for scene recognition and UCF101(Soomro et al., 2012) for human motion analysis, which are critical in evaluating the effectiveness of our proposed methods.

### 4.2 BASELINE SETTINGS

**Base-to-novel class generalization:** In this scenario, we adhere to the conventional standard of dividing the datasets into base and novel classes. The model is trained exclusively on the base classes, yet the fine-tuned model is evaluated on both the base and novel classes. This experiment demonstrates the proposed methods' capabilities for model generalization.

**Cross-dataset evaluation:** In this evaluation, our model is trained exclusively on the ImageNet dataset and then tested on other datasets without any fine-tuning.

**Few-shot learning:** We test our model's generalization for different K shots per class on each dataset, where K = 1, 2, 4, 8, 16.

### 4.3 BASE-TO-NOVEL GENERALIZATION

We compared the performance of our proposed methods with previous approaches such as those reported by (Zhou et al., 2022a), (Lu et al., 2022), MaPLe (Khattak et al., 2023a), and previous SOTA methods described in PromptSRC (Khattak et al., 2023b). As shown in Tab. 1, our model outperforms all other methods in terms of both base and novel class performance. Notably, our methods demonstrate significantly faster convergence speeds, surpassing all previous SOTA results by epoch 10, which is twice as fast. Moreover, at 5 epochs, DPD-LoRA exhibits better generalization and

Table 1: **Comparative Analysis of Base-to-Novel Generalization Performance of DPD-LoRA**. DPD-LoRA shows consistent improvement over previous SOTA methods. Note that † refers to results reproduced using the official code under a identical device and configuration to ours. The best accuracy is highlighted in bold. section 4.3

| Dataset | | CLIP | CoOp | CoCoOp | ProDA | MaPLe | MaPLe† | PromptSRC | PromptSRC† | ALIGN | Ours |
|---|---|---|---|---|---|---|---|---|---|---|---|
| Avg. on 11 datasets | Base | 69.34 | 82.69 | 80.47 | 81.56 | 82.28 | 82.31 | 84.26 | 84.24 | 83.38 | **85.74** |
| | Novel | 74.22 | 63.22 | 71.69 | 72.30 | 75.14 | 74.06 | 76.10 | 76.02 | 75.51 | **76.89** |
| | HM | 71.70 | 71.66 | 75.83 | 76.65 | 78.55 | 77.97 | 79.97 | 79.92 | 79.25 | **81.07** |
| ImageNet | Base | 72.43 | 76.47 | 75.98 | 75.40 | 76.66 | 76.77 | 77.60 | 77.67 | 76.89 | **78.13** |
| | Novel | 68.14 | 67.88 | 70.43 | 70.23 | 70.54 | 70.8 | 70.73 | 70.37 | **72.15** | 71.33 |
| | HM | 70.22 | 71.92 | 73.10 | 72.72 | 73.47 | 73.66 | 74.01 | 73.84 | 74.45 | **74.58** |
| Caltech101 | Base | 96.84 | 98.00 | 97.96 | 98.27 | 97.74 | 98.03 | 98.10 | 98.07 | 98.37 | **98.53** |
| | Novel | 94.00 | 89.81 | 93.81 | 93.23 | 94.36 | 94.57 | 94.03 | 94.10 | 94.70 | **94.77** |
| | HM | 95.40 | 93.73 | 95.84 | 95.68 | 96.02 | 96.27 | 96.02 | 96.04 | 96.50 | **96.61** |
| OxfordPets | Base | 91.17 | 93.67 | 95.20 | 95.43 | 95.43 | 95.2 | 95.33 | 95.33 | 95.67 | **95.77** |
| | Novel | 97.26 | 95.29 | 97.69 | 97.83 | 97.76 | 97.63 | 97.30 | 97.27 | **97.93** | 97.90 |
| | HM | 94.12 | 94.47 | 96.43 | 96.62 | 96.58 | 96.4 | 96.30 | 96.29 | 96.79 | **96.86** |
| Stanford Cars | Base | 63.37 | 78.12 | 70.49 | 74.70 | 72.94 | 72.63 | 78.27 | 78.27 | 77.24 | **83.57** |
| | Novel | 74.89 | 60.40 | 73.59 | 71.20 | 74.00 | 73.90 | 74.97 | 75.30 | **76.38** | 75.30 |
| | HM | 68.65 | 68.13 | 72.01 | 72.91 | 73.47 | 73.26 | 76.58 | 76.76 | 76.80 | **79.22** |
| Flowers102 | Base | 72.08 | 97.60 | 94.87 | 97.70 | 95.92 | 96.2 | 98.07 | 98.03 | 97.70 | **98.37** |
| | Novel | **77.80** | 59.67 | 71.75 | 68.68 | 72.46 | 71.73 | 76.50 | 76.83 | 73.3 | 76.83 |
| | HM | 74.83 | 74.06 | 81.71 | 80.66 | 82.56 | 82.18 | 85.95 | 86.14 | 83.75 | **86.28** |
| Food101 | Base | 90.10 | 88.33 | 90.70 | 90.30 | 90.71 | 90.57 | 90.67 | 90.67 | **90.77** | 90.00 |
| | Novel | 91.22 | 82.26 | 91.29 | 88.57 | 92.05 | 92.07 | 91.53 | 91.47 | **92.07** | 92.03 |
| | HM | 90.66 | 85.19 | 90.99 | 89.43 | 91.38 | 90.55 | 91.10 | 91.07 | **91.42** | 91.00 |
| FGVC Aircraft | Base | 27.19 | 40.44 | 33.41 | 36.90 | 37.44 | 37.63 | 42.73 | 42.77 | 37.56 | **49.93** |
| | Novel | 36.29 | 22.30 | 23.71 | 34.13 | 35.61 | 35.60 | **37.87** | 36.60 | 36.97 | 36.03 |
| | HM | 31.09 | 28.75 | 27.74 | 35.46 | 36.50 | 36.59 | 40.15 | 39.45 | 37.26 | **41.86** |
| SUN397 | Base | 69.36 | 80.60 | 79.74 | 78.67 | 80.82 | 80.83 | **82.67** | **82.67** | 82.47 | 81.80 |
| | Novel | 75.35 | 65.89 | 76.86 | 76.93 | 78.80 | 77.77 | 78.47 | 78.33 | **79.68** | 79.33 |
| | HM | 72.23 | 72.51 | 78.27 | 77.79 | 79.75 | 79.27 | 80.52 | 80.44 | **81.05** | 80.55 |
| DTD | Base | 53.24 | 79.44 | 77.01 | 80.67 | 80.36 | 80.23 | **83.37** | 83.30 | 82.13 | 83.83 |
| | Novel | 59.90 | 41.18 | 56.00 | 56.48 | 59.18 | 55.03 | 62.97 | 62.50 | 54.17 | **63.97** |
| | HM | 56.37 | 54.24 | 64.85 | 66.44 | 68.16 | 65.28 | 71.75 | 71.42 | 65.28 | **72.57** |
| Eurosat | Base | 56.48 | 92.19 | 87.49 | 83.90 | 94.07 | 93.4 | 92.90 | 92.70 | 94.03 | **95.47** |
| | Novel | 64.05 | 54.74 | 60.04 | 66.00 | 73.23 | 69.83 | 73.90 | 74.80 | 74.90 | **79.43** |
| | HM | 60.03 | 68.69 | 71.21 | 73.88 | 82.35 | 79.91 | 82.32 | 82.79 | 83.38 | **86.71** |
| UCF101 | Base | 70.53 | 84.69 | 82.33 | 85.23 | 83.00 | 83.97 | **87.73** | 87.20 | 84.43 | **87.73** |
| | Novel | 77.50 | 56.05 | 73.45 | 71.97 | 78.66 | 77.30 | 78.80 | 78.60 | 78.33 | **78.83** |
| | HM | 73.85 | 67.46 | 77.64 | 78.04 | 80.77 | 80.5 | 82.74 | 82.68 | 81.27 | **83.04** |

harmonic mean scores than the SOTA PromptSRC at 20 epochs. Specifically, under the same training duration as our baseline MaPLe, DPD-LoRA achieved a 4.17% increase in base performance while enhancing novel class performance by 3.82%. Conversely, MaPLe's training period ended at epoch 5 due to an extremely overfitting trend. However, with our proposed Self-Regularized DPD-

Table 2: **Comparative Assessment of DPD-LoRA in Cross-Dataset Evaluation.** Prompt learning methods are trained on ImageNet and evaluated on cross-datasets. Note that † refers to results reproduced using the official code under a identical device and configuration to ours. The best accuracy is highlighted in bold. section 4.4

| | Caltech | Pets | Cars | Flowers | Food101 | Aircraft | SUN397 | DTD | EuroSAT | UCF101 | *Average* |
|---|---|---|---|---|---|---|---|---|---|---|---|
| CLIP | 93.35 | 88.25 | 65.48 | 67.44 | 83.65 | 23.67 | 62.59 | 44.27 | 42.01 | 65.13 | 63.58 |
| CoOp | 93.70 | 89.14 | 64.51 | 68.71 | 85.30 | 18.47 | 64.15 | 41.92 | 46.39 | 66.55 | 63.88 |
| CoCoOp | 93.79 | 90.46 | 64.90 | 70.85 | 83.97 | 22.29 | 66.89 | 45.45 | 39.23 | 68.44 | 64.63 |
| MaPLe† | 93.80 | 90.23 | **66.17** | 71.57 | 86.33 | 23.93 | 67.53 | 45.50 | 43.63 | 68.23 | 65.69 |
| PromptSRC† | 93.40 | 90.30 | 65.30 | 70.63 | 86.30 | 24.17 | 67.03 | **47.27** | 44.67 | **69.00** | 65.80 |
| DPD-LoRA(Ours) | **93.97** | **90.47** | 66.00 | **71.93** | **86.40** | **24.50** | **67.57** | 46.43 | **46.93** | 68.40 | **66.25** |

LoRA, there is continued learning beyond this point. Although PromptSRC addressed this issue by extending the training phase to 20 epochs, our proposed method still surpasses them by epoch 10 and achieves the best performance at epoch 20.

## 4.4 CROSS DATASET EVALUATION

We compare the cross-dataset performance of our method with previous methods, as shown in Tab. 2. Our method demonstrates better generalization than CoOp and CoCoOp on 10 out of 10 datasets. Compared to our baseline, MaPLe, we also achieve superior performance on 9 out of 10 datasets. Finally, compared to the previous state-of-the-art method, PromptSRC, we show improved performance on 8 out of 10 datasets. Overall, this results in better generalization on average.

Figure 4: **Few-shot Performance Evaluation:** DPD-LoRA with the ViT/16 CLIP backbone demonstrates comparable performance in few-shot experiments, achieving the highest overall performance gain across all shots for 11 datasets. section 4.5

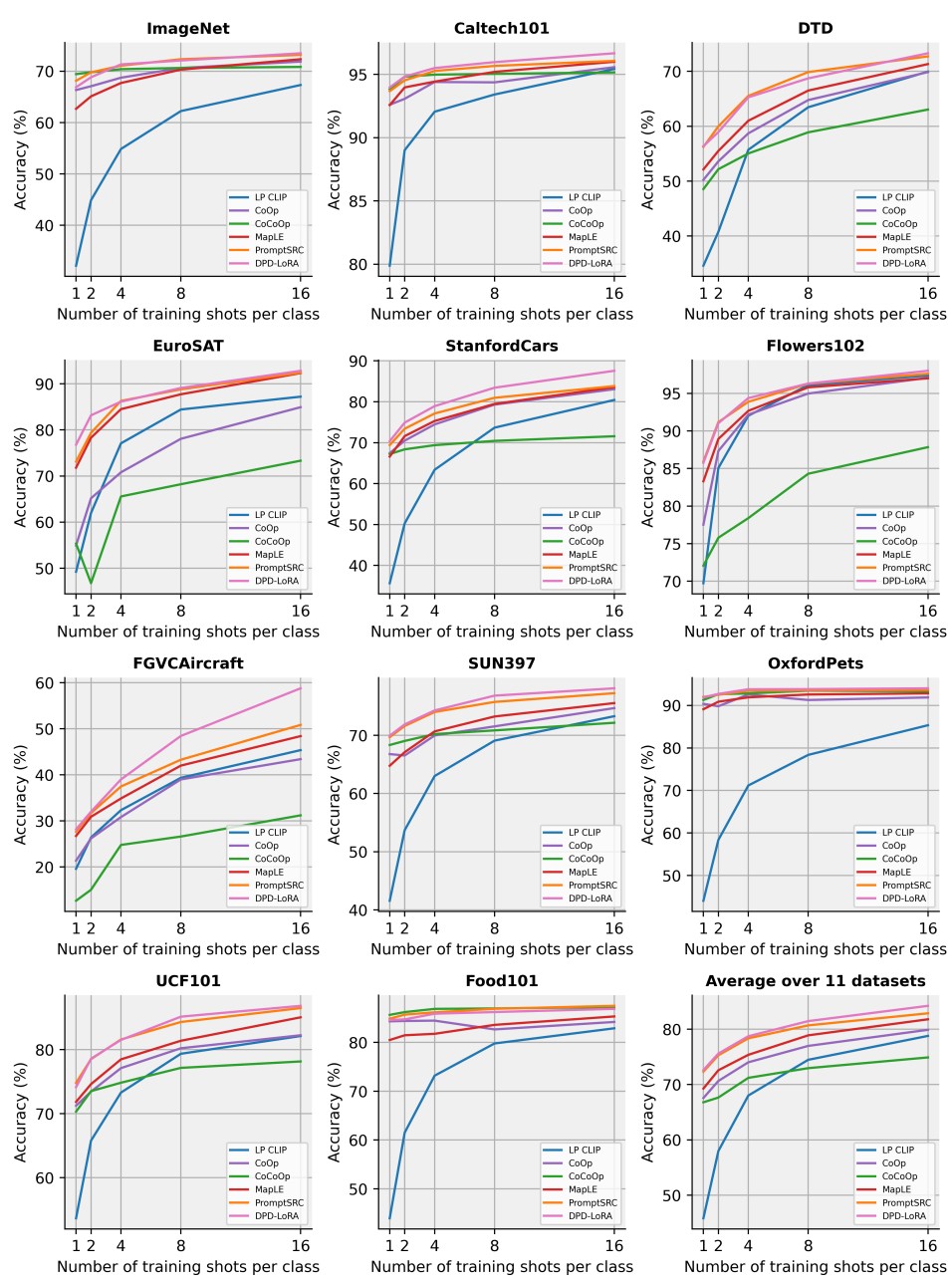

| Method | Base Acc. | Novel Acc. | HM |
|---|---|---|---|
| Baseline | 82.31 | 74.06 | 77.97 |
| + LoRSS | 85.36 | 75.85 | 80.32 |
| + Interaction | 85.35 | 76.33 | 80.59 |
| + $\mathcal{L}_{\text{SCL-LoRA}}$ | 85.17 | 76.73 | 80.73 |
| + PCGM | **85.74** | **76.89** | **81.07** |

Table 3: **Effectiveness of DPD-LoRA:** Effect of the proposed components in our model. The results are averaged over 11 datasets, with 'HM' denoting the harmonic mean of Base and Novel. The effectiveness of each component is evaluated at epoch 20. section 4.6

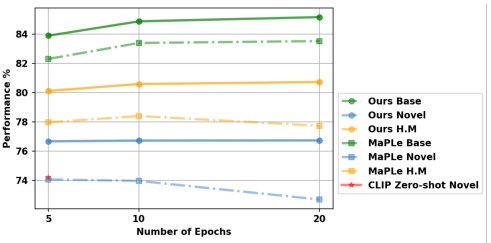

Figure 5: **DPD-LoRA v.s. MaPLe:** DPD-LoRA exhibits a more stable generalization and less forgetting with an increase in the number of training epochs. section 3.5

## 4.5 FEW-SHOT EXPERIMENTS

We conduct a comparison with previous methods in a few-shot setting, as shown in Fig. 4. Notably, our method has achieved better zero-shot performance (base-to-novel experiments). To further evaluate its effectiveness, we test whether our method can maintain this superior performance in a few-shot learning scenario. For concrete results, please refer to our appendix Tab. 6. We found that for all shots, our method shows superior performance.

## 4.6 ABLATION STUDY

**Effectiveness of different components of DPD-LoRA:** We conducted an ablation study to assess the effectiveness of our methods on accuracy and generalization across 11 datasets. By isolating different components and excluding other factors, we evaluated their individual contributions to the improvements observed in Base-to-Novel experiments, as shown in Tab. 3. Our findings demonstrate that incorporating these components significantly enhances the pretrained model's generalization. The optimal performance was achieved by combining all components, affirming the effectiveness of our methods. While more training epochs improved base class performance, they often harmed novel class accuracy. Although our first proposed component, LoRSS, outperformed the SOTA in HM and base metrics, its novel performance remained suboptimal. To address this, we introduced a self-constrain loss on LoRSS to balance base and novel performance. Additionally, our PCGM further enhances adaptation and generalization capabilities.

We conducted an ablation study to assess the effectiveness of our methods on accuracy and generalization across 11 datasets. By isolating different components and excluding other factors, we evaluated their individual contributions to the improvements observed in Base-to-Novel experiments, as shown in Tab. 3. Our findings demonstrate that incorporating these components significantly enhances the pretrained model's generalization. The optimal performance was achieved by combining all components, affirming the effectiveness of our methods. While more training epochs improved base class performance, they often harmed novel class accuracy. Although our first proposed component, LoRSS, outperformed the SOTA in HM and base metrics, its novel performance remained suboptimal. To address this, we introduced a self-constrain loss on LoRSS to balance base and novel performance. Additionally, our PCGM further enhances adaptation and generalization capabilities.

**Generalization ability of DPD-LoRA:** Furthermore, we compare our method with the baseline MaPLe(Khattak et al., 2023a), increasing the number of epochs. It is evident that MaPLe's performance declines. In contrast, our proposed methods remain robust in both adaptation and generalization performance, as shown in Fig. 5.

## 5 CONCLUSION

This paper addresses the limitations of traditional parameter-efficient fine-tuning and prompt learning methods by proposing a novel approach that integrates prompt learning to guide LoRA learning distribution. We introduced a prompt-steered LoRA method, developed a self-adaptive loss strategy to enhance lower-rank subspaces distribution, and reused deep prompt vectors to provide comprehensive guidance for LoRA layers. Our work not only advances the field of parameter-efficient fine-

tuning but also opens new avenues for leveraging prompts to enhance model performance across various downstream tasks.

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

## A  IMPLEMENTATION DETAILS

### A.1  HARDWARE AND SOFTWARE DETAILS

We implemented our method using the MaPLe (Khattak et al., 2023a) and PromptSRC(Khattak et al., 2023b) with the public available CLIP ViT-B/16 backbone architecture. Our models were trained on a single NVIDIA H100 GPU for all evaluation, i.e. base-to-novel, cross-dataset and few shots benchmarks, using the PyTorch framework.

### A.2  DPD-LoRA TRAINING DETAILS

In this section, we provide the settings used to train each of the different tasks. And DPD-LoRA is described in **Algorithm 1**.

**Base-to-novel benchmark**: We follow previous methods, using a deep prompt token (d=9) and a fixed learning rate of 0.005 for both prompt tokens and PCGM parameters, with the learning rate for LoRA set to 0.0035. We apply early stopping at epoch 5 for all settings of 5, 10, and 20 epochs. Notably, our $\mathcal{L}_{\text{SCL-LoRA}}$ weight factor $\lambda$ is fixed at a value of 0.1.

**Cross-dataset benchmark**: In Tab. 2 of the main paper, we compare the performance of DPD-LoRA with current state-of-the-art prompt learning methods, MaPLe and PromptSRC, for cross-dataset generalization. Notably, our methods only train for 2 epochs in this evaluation(fowlloing MaPLe), making $\mathcal{L}_{\text{SCL-LoRA}}$ inapplicable in this scenario. Thus, in cross-dataset scenarios, we set the factor $\lambda$ to zero.

**Hyper-parameters**: Table 4a shows the hyper-parameters used to train the DPD-LoRA model within different evaluations. We minimize our learning parameters by using a fixed rank(r) of 12 for all plain LoRA settings. Under the same parameters, we use a fixed rank(r) of 4 and m=3 for all LoRSS configurations.

**Complexity Analysis**: Tab. 4b provides the computational analysis. We aconclude that, even with fewer parameters, our model provides better adaptation and generalization capabilities.

| Hyper parameter | Base-to-Novel | Cross dataset | Few-shots |
|---|---|---|---|
| LoRA LR | 0.035 | 0.02 | 0.035 |
| Prompt token LR | 0.05 | 0.035 | 0.035 |
| PCGM LR | 0.05 | 0.035 | 0.035 |
| Epochs | 20 | 2 | 50 |
| $\alpha$ | 0.9 | 0.9 | 0.9 |
| $r/m$ | 4/3 | 4/3 | 4/3 |
| Early-Stop | 5 | - | 20 |
| $\lambda_{orth}$ | 0.1 | 0.1 | 0.1 |
| $\lambda_{lora-scl}$ | 0.1 | - | 0.01 |

(a) Hyper-parameter settings used in different techniques for various benchmark settings.

| Method | Params | Params % CLIP | Base | Novel | HM | FPS |
|---|---|---|---|---|---|---|
| CoOp | 2048 | 0.002 | 82.69 | 63.22 | 71.66 | 104.5 |
| CoCoOp | 35360 | 0.03 | 80.47 | 71.69 | 75.83 | 53.3 |
| Independent V-L | 31488 | 0.02 | 82.15 | 74.07 | 77.90 | 149.86 |
| MaPLe | 3.55 M | 2.85 | 82.28 | 75.14 | 78.55 | 175.58 |
| ALIGN | 3.58 M | 2.87 | 83.38 | 75.51 | 79.25 | 72.6 |
| **DPD-LoRA†** | **1.92 M** | **1.54** | **84.80** | **76.80** | **80.60** | 82.5 |
| DPD-LoRA | 4.72 M | 3.79 | 85.67 | 76.91 | 81.05 | 81.57 |

(b) Comparison of computational complexity among different prompting methods. † denote a degraded version of DPD-LoRA that utilizes the same MLP, which projects textual prompts to visual prompts across all layers.

Table 4: Overview of hyper-parameter settings and computational complexity comparisons among different methods.

## B  FEW-SHOT PERFORMANCE EVALUATION:

Table 6 shows the detailed performance evaluation on the different few-shot settings.

| Model | Base(%)↑ | Novel(%)↑ | HM ↑ |
|---|---|---|---|
| **Baseline/Naive combinations** | | | |
| CLIP | 72.43 | 68.14 | 70.22 |
| plain LoRA$_{(x)}$ (5 epoch) | 77.57 | 69.70 | 73.42 |
| Prompts Learning (MaPLe)$_{(x')}$ (5 epoch) | 76.77 | 70.80 | 73.66 |
| Frozen LoRA$_{(x)}$ + Prompts$_{(x')}$ (5 + 5 epoch) | 75.20 | 61.17 | 67.46 |
| Frozen Prompts$_{(x')}$ + LoRA$_{(x')}$ (5 + 5 epoch) | 76.77 | 70.47 | 73.49 |
| **Prompt-Driven Adaptation** | | | |
| Prompt-Driven plain LoRA (5 epoch) | 77.62 | 70.81 | 74.09 |
| Prompt-Driven LoRSS (5 epoch) | 77.63 | 70.97 | 74.15 |
| **Ours** | | | |
| DPD-LoRA (full model) (5 epoch) | 77.87 | 71.13 | 74.34 |
| DPD-LoRA (full model) (20 epoch) | **78.13** | **71.33** | **74.58** |

Table 5: Ablation experiments for Prompts-To-LoRA on the ImageNet dataset. Here, $x$ refers to the original input, while $x'$ denotes the prompted input (i.e., the concatenation of $x$ and prompt tokens). Note that the plain LoRA here is distinct from our proposed LoRSS. Only the last two rows represent the performance of our full model.

## C  ADDITIONAL ABLATION STUDY

## D  MATHEMATICAL DERIVATION

In the main paper, we discussed the integration of **Prompt Learning** and **Low-Rank Adaptation (LoRA)** in transformers. Here, we delve deeper into their combined effect on the Multi-Head Attention (MHA) mechanism.

### D.1  PROMPT LEARNING IN TRANSFORMERS

Prompt Learning involves introducing a set of learnable tokens, which are appended to the original input. This can be implemented in the text branch with **Textual Prompts** $P_\text{t} = \{p_\text{t}^1, p_\text{t}^2, \ldots, p_\text{t}^{n_p}\}$ (Zhou et al., 2022b), or in the visual branch with **Visual Prompts** $P_\text{v} = \{p_\text{v}^1, p_\text{v}^2, \ldots, p_\text{v}^{n_p}\}$ (Jia et al., 2022), or simultaneously in both branches (Khattak et al., 2023a). The new inputs, incorporating these prompts, replace the original text and visual inputs, yielding:

$$\widetilde{Y}_p = [t_\text{sos},\, P_\text{t},\, t_1,\, t_2,\, \ldots,\, t_L,\, t_\text{cls},\, t_\text{eos}],$$
$$\widetilde{X}_p = [P_\text{v},\, e_1,\, e_2,\, \ldots,\, e_N,\, e_\text{cls}], \tag{11}$$

where $[\,\cdot\,]$ denotes concatenation, $t_i$ are text tokens, and $e_i$ are visual embeddings.

### D.2  MULTI-HEAD ATTENTION MECHANISM

We recall the interaction of these prompted inputs with the **Multi-Head Attention (MHA)** mechanism in the transformer architecture:

$$\text{MHA}(X) = \text{Concat}(\text{head}_1, \ldots, \text{head}_h)\, W^O,$$
$$\text{where} \quad \text{head}_i = \text{Attention}(Q_i, K_i, V_i) = \phi\left(\frac{Q_i K_i^\top}{\sqrt{d_k}}\right) V_i, \tag{12}$$
$$Q_i = X' W_i^Q, \quad K_i = X' W_i^K, \quad V_i = X' W_i^V.$$

In this formulation, $\phi$ represents the softmax function, $h$ is the number of attention heads, $d_k$ is the dimension of the key vectors, and $X'$ refers to the prompted inputs $\widetilde{Y}_p$ or $\widetilde{X}_p$.

### D.3  EXPANSION WITH PROMPTED INPUTS

To illustrate the effect of prompts more clearly, we simplify the attention head computation by omitting the scalar $\sqrt{d_k}$:

$$\text{head}_i(X') = \phi(Q_i K_i^\top) V_i. \tag{13}$$

We expand the terms for the prompted inputs, considering the pretrained weights $W_Q, W_K, W_V$, and write them as:

$$Q' = X'W_Q = [X; P]W_Q = \begin{bmatrix} X \\ P \end{bmatrix} W_Q = \begin{bmatrix} XW_Q \\ PW_Q \end{bmatrix},$$

$$K' = X'W_K = [X; P]W_K = \begin{bmatrix} X \\ P \end{bmatrix} W_K = \begin{bmatrix} XW_K \\ PW_K \end{bmatrix}, \tag{14}$$

$$V' = X'W_V = [X; P]W_V = \begin{bmatrix} X \\ P \end{bmatrix} W_V = \begin{bmatrix} XW_V \\ PW_V \end{bmatrix}.$$

The attention mechanism with prompted inputs becomes:

$$\text{Attn}(Q', K', V') = \phi(Q'K'^\top)V'$$

$$= \phi\left( \begin{bmatrix} XW_Q \\ PW_Q \end{bmatrix} \begin{bmatrix} XW_K \\ PW_K \end{bmatrix}^\top \right) \begin{bmatrix} XW_V \\ PW_V \end{bmatrix}. \tag{15}$$

This expands to:

$$\text{Attn}(Q', K', V') = \begin{bmatrix} \phi(XW_Q(XW_K)^\top)XW_V + \phi(XW_Q(PW_K)^\top)PW_V \\ \phi(PW_Q(XW_K)^\top)XW_V + \phi(PW_Q(PW_K)^\top)PW_V \end{bmatrix}. \tag{16}$$

### D.4 INCORPORATING LOW-RANK ADAPTATION (LoRA)

To incorporate LoRA into this framework, we adapt the weight matrices by adding low-rank updates:

$$\begin{aligned}
W_Q &= W_Q^{\text{base}} + \Delta W_Q, \quad \Delta W_Q = A_Q B_Q, \\
W_K &= W_K^{\text{base}} + \Delta W_K, \quad \Delta W_K = A_K B_K, \\
W_V &= W_V^{\text{base}} + \Delta W_V, \quad \Delta W_V = A_V B_V, \\
W^O &= W^{O,\text{base}} + \Delta W^O, \quad \Delta W^O = A^O B^O,
\end{aligned} \tag{17}$$

where $A_* \in \mathbb{R}^{d_{\text{model}} \times r}$ and $B_* \in \mathbb{R}^{r \times d_k}$ are learnable matrices with $r \ll d_{\text{model}}$. The low-rank matrices enable efficient fine-tuning by reducing the number of trainable parameters.

Substituting the adapted weights into the attention computation, we have:

$$\begin{aligned}
Q' &= X'(W_Q^{\text{base}} + A_Q B_Q) = X'W_Q^{\text{base}} + X'A_Q B_Q, \\
K' &= X'W_K^{\text{base}} + X'A_K B_K, \\
V' &= X'W_V^{\text{base}} + X'A_V B_V.
\end{aligned} \tag{18}$$

The attention mechanism with LoRA and prompted inputs becomes:

$$\text{Attn}_{\text{LoRA}}(Q', K', V') = \phi(Q'K'^\top)V' =$$

$$\begin{bmatrix} \phi\left( XW_Q^{\text{total}}(XW_K^{\text{total}})^\top \right) XW_V^{\text{total}} + \phi\left( XW_Q^{\text{total}}(PW_K^{\text{total}})^\top \right) PW_V^{\text{total}} \\ \phi\left( PW_Q^{\text{total}}(XW_K^{\text{total}})^\top \right) XW_V^{\text{total}} + \phi\left( PW_Q^{\text{total}}(PW_K^{\text{total}})^\top \right) PW_V^{\text{total}} \end{bmatrix}, \tag{19}$$

where $W_Q^{\text{total}} = W_Q^{\text{base}} + A_Q B_Q$, and similarly for $W_K^{\text{total}}$ and $W_V^{\text{total}}$.

### D.5 INTERPRETATION

The introduction of the prompt $P$ into the LoRA mechanism does not alter the fundamental role or operation of the low-rank matrices $A$ and $B$. Instead, it extends their application scope by adding new inputs for the model to handle. Specifically, $A$ and $B$ continue to adjust the weights $W_Q$, $W_K$, and $W_V$ through low-rank updates. However, with $P$'s incorporation, these matrices now accommodate both the original input $X$ and the additional information provided by $P$.

Our experimental results (see Tab. 5) indicate that prompts contribute significantly to the LoRA layer. The bolded rows show that prompts provide the low-rank matrices $A$ and $B$ with additional information to guide the weight updates, enhancing the model's performance.

## E  VISUALIZATION

To demonstrate the superiority of our proposed DPD-LoRA method over the baseline MaPLe, we visualize and compare the image embeddings generated by both methods using t-SNE (Fig 6). t-SNE provides a two-dimensional representation of high-dimensional data. The visualization shows that the image embeddings of DPD-LoRA are more separable, indicating that the Dynamic Prompt-Driven Low-Rank Adaptation contributes to improved performance of the image encoder. We chose the EuroSAT dataset for demonstration; among the 11 datasets, it has fewer classes, making it easier to observe the model's generalization capability.

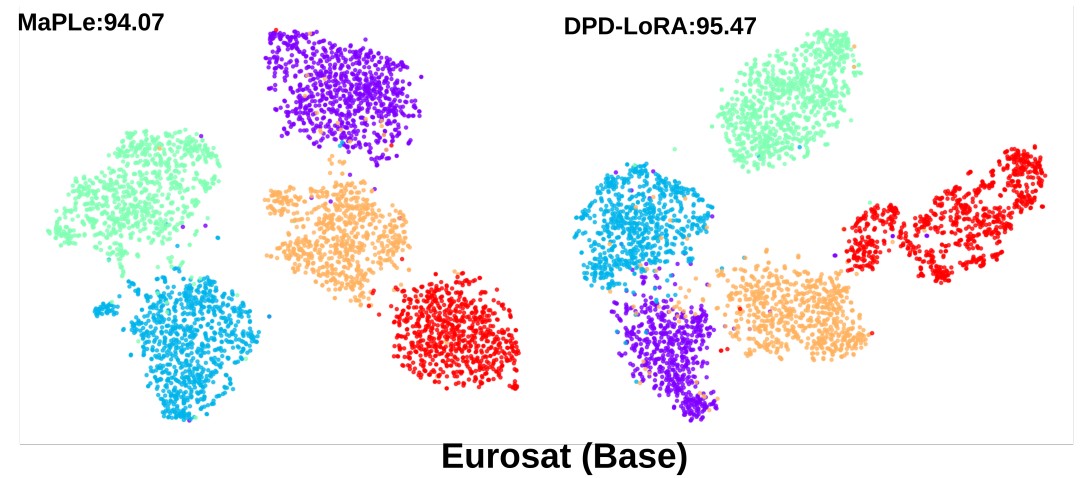

**Eurosat (Base)**

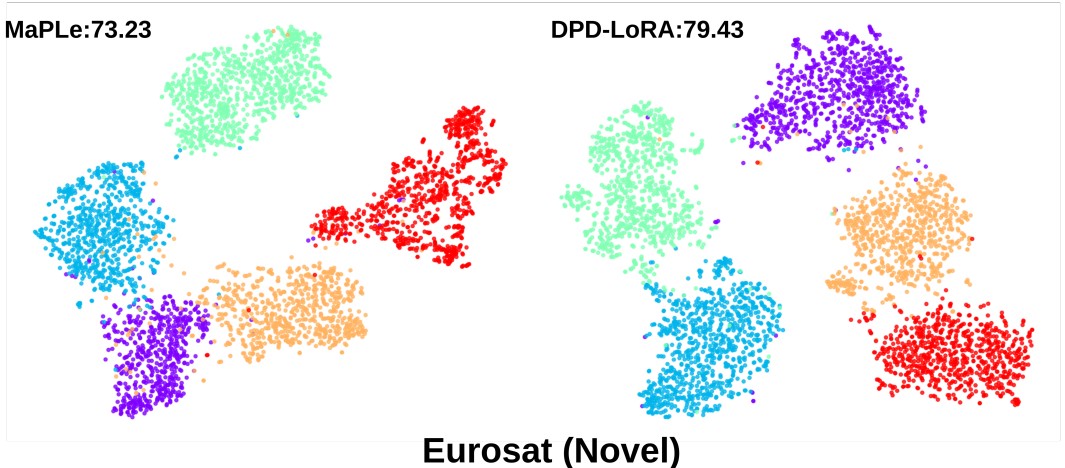

**Eurosat (Novel)**

Figure 6: **Visualization of Image Embeddings.** t-SNE plots of image embeddings generated by our method (DPD-LoRA) and the baseline (MaPLe) on the EuroSAT (Novel) dataset. The visualization demonstrates how our method separates different classes.

---

**Algorithm 1** DPD-LoRA

---

**Input:** Dataset $\mathcal{D} = \{X, y\}^N$, CLIP Model $\theta_{\text{CLIP}} = \{\theta_g, \theta_f\}$
**Stage 1:** Training a LoRSS-CLIP model.
**Initialize:** LoRA matrices $A_l$, $B_l$ for layers $l$ in $\theta_f$ and $\theta_g$. Number (m) of low-rank matrices = 3. Rank (r) of each low-rank matrices r = 4.
**for** $i \in [1, early_{stop}^T]$ **do**
    sample data $\{X, y\} \subseteq \mathcal{D}$
    `// Apply LoRA updates for the iteration`
    `// Encourage hierarchical interacting with its preceding layer`
    **if** layer$(l) > 0$ **then**
        $\Delta w \leftarrow \beta(A_l B_l) + \gamma(A_{(l-1)} B_{(l-1)})$
    **else**
        $\Delta w \leftarrow (A_l B_l)$
    **end if**
    Update $\theta_f \leftarrow \theta_f^l + \Delta w^f$ for each layer $l$ in $\theta_f$
    Update $\theta_g \leftarrow \theta_g^l + \Delta w^g$ for each layer $l$ in $\theta_g$
    $\boldsymbol{f} \leftarrow f(\boldsymbol{x}, \theta_f), \boldsymbol{g} \leftarrow g(\boldsymbol{y}, \theta_g)$
    `// normal cross-entropy (CE) supervision loss.`
    $\mathcal{L}_{\text{total}} \leftarrow \mathcal{L}_{\text{CE}}(\text{sim}(\boldsymbol{f}, \boldsymbol{g}), y) + \mathcal{L}_{\text{orth}} + \mathcal{L}_{\text{textual-diversity}}$
    `// update all LoRA matrices` $A_l$, $B_l$ `(LoRSS distribution) with`
    `supervision loss.`
    $\boldsymbol{w}_{t+1} \leftarrow \boldsymbol{w}_t - \eta \nabla_{\boldsymbol{w}_t} \mathcal{L}_{\text{total}}$
**end for**

**Stage 2:** Training a DPD-LoRA CLIP model.
**Initialize:** LoRA matrices $A_l$, $B_l$ for layers $l$ in $\theta_f$ and $\theta_g$. Prompt vectors $\boldsymbol{P} = \{\boldsymbol{P_v}, \boldsymbol{P_t}\}$. Number (m) of low-rank matrices = 3. Rank (r) of each low-rank matrices r = 4.
**for** $i \in [1, T]$ **do**
    sample data $\{X, y\} \subseteq \mathcal{D}$
    `// Hierarchical interaction among deep prompts, and LoRA layers`
    **if** layer$(l) > 0$ **then**
        $\boldsymbol{P} \leftarrow \alpha(P_l) + (1 - \alpha) P_{l-1}$
        $\Delta w \leftarrow \alpha(A_l B_l) + (1 - \alpha)(A_{(l-1)} B_{(l-1)})$
    **else**
        $\boldsymbol{P} \leftarrow (P_l)$
        $\Delta w \leftarrow (A_l B_l)$
    **end if**
    `// apply PCGM`
    $\Delta w \leftarrow \Delta w \mathbf{G}(\mathbf{P})$
    `// Using` $\theta_{\text{CLIP}}$ `with` $\boldsymbol{P}$ `and` $\boldsymbol{\Delta w}$, `obtain prompted LoRA visual and text`
    `features`
    $\tilde{\boldsymbol{f}}_{\boldsymbol{p}} \leftarrow f(\tilde{\boldsymbol{x}}_{\boldsymbol{p}}, \theta_f + \Delta w_f), \tilde{\boldsymbol{g}}_{\boldsymbol{p}} \leftarrow g(\tilde{\boldsymbol{y}}_{\boldsymbol{p}}, \theta_g + \Delta w_g)$
    `// normal CE supervision loss to learning prompted LoRSS distribution.`
    $\mathcal{L}_{\text{sup}} \leftarrow \mathcal{L}_{\text{CE}}(\text{sim}(\tilde{\boldsymbol{f}}_{\boldsymbol{p}}, \tilde{\boldsymbol{g}}_{\boldsymbol{p}}), y)$
    Obtain pre-trained fixed lora distribution $D_{\text{lorss}}$
    `// KL divergence for two distributions self-regularizing loss.`
    $\mathcal{L}_{\text{SCL}} \leftarrow \mathcal{L}_{\text{KL}}(\Delta w(D_{\text{lorss}}) || \Delta w(D_{\text{DPD-LoRA}})) + \mathcal{L}_{\text{orth}}$
    $\mathcal{L}_{\text{final}} \leftarrow \mathcal{L}_{\text{sup}} + \lambda \mathcal{L}_{\text{SCL}} + \mathcal{L}_{\text{textual-diversity}}$
    `// update prompt vectors and DPD-LoRA distribution with combined loss.`
    $\boldsymbol{P}_{t+1} \leftarrow \boldsymbol{P}_t - \eta \nabla_{\boldsymbol{P}_t} \mathcal{L}_{\text{final}}$
    $\boldsymbol{w}(\boldsymbol{D}_{\text{DPD-LoRA}})_{t+1} \leftarrow \boldsymbol{w}(\boldsymbol{D}_{\text{DPD-LoRA}})_t - \eta \nabla_{\boldsymbol{\Delta w}(\boldsymbol{D}_{\text{DPD-LoRA}})_t} \mathcal{L}_{\text{final}}$
**end for**

---

Table 6: **Comparative Assessment of DPD-LoRA on Few-shot Evaluation:** Per-dataset performance comparison of DPD-LoRA with various methods in few-shot setting.

| Dataset | Method | 1 shot | 2 shots | 4 shots | 8 shots | 16 shots |
|---------|--------|--------|---------|---------|---------|----------|
| ImageNet | Linear probe CLIP | 32.13 | 44.88 | 54.85 | 62.23 | 67.31 |
| | CoOp | 66.33 | 67.07 | 68.73 | 70.63 | 71.87 |
| | CoCoOp | 69.43 | 69.78 | 70.39 | 70.63 | 70.83 |
| | MaPLe | 62.67 | 65.10 | 67.70 | 70.30 | 72.33 |
| | PromptSRC | 68.13 | 69.77 | 71.07 | 72.33 | 73.17 |
| | DPD-LoRA (Ours) | 65.87 | 68.82 | 71.32 | 72.08 | 73.49 |
| Caltech101 | Linear probe CLIP | 79.88 | 89.01 | 92.05 | 93.41 | 95.43 |
| | CoOp | 92.60 | 93.07 | 94.40 | 94.37 | 95.57 |
| | CoCoOp | 93.83 | 94.82 | 94.98 | 95.04 | 95.16 |
| | MaPLe | 92.57 | 93.97 | 94.43 | 95.20 | 96.00 |
| | PromptSRC | 93.67 | 94.53 | 95.27 | 95.67 | 96.07 |
| | DPD-LoRA (Ours) | 93.97 | 94.83 | 95.50 | 95.97 | 96.67 |
| DTD | Linear probe CLIP | 34.59 | 40.76 | 55.71 | 63.46 | 69.96 |
| | CoOp | 50.23 | 53.60 | 58.70 | 64.77 | 69.87 |
| | CoCoOp | 48.54 | 52.17 | 55.04 | 58.89 | 63.04 |
| | MaPLe | 52.13 | 55.50 | 61.00 | 66.50 | 71.33 |
| | PromptSRC | 56.23 | 59.97 | 65.53 | 69.87 | 72.73 |
| | DPD-LoRA (Ours) | 56.37 | 58.93 | 65.27 | 68.73 | 73.30 |
| EuroSAT | Linear probe CLIP | 49.23 | 61.98 | 77.09 | 84.43 | 87.21 |
| | CoOp | 54.93 | 65.17 | 70.80 | 78.07 | 84.93 |
| | CoCoOp | 55.33 | 46.74 | 65.56 | 68.21 | 73.32 |
| | MaPLe | 71.80 | 78.30 | 84.50 | 87.73 | 92.33 |
| | PromptSRC | 73.13 | 79.37 | 86.30 | 88.80 | 92.43 |
| | DPD-LoRA (Ours) | 76.8 | 83.17 | 86.13 | 89.1 | 92.83 |
| StanfordCars | Linear probe CLIP | 35.66 | 50.28 | 63.38 | 73.67 | 80.44 |
| | CoOp | 67.43 | 70.50 | 74.47 | 79.30 | 83.07 |
| | CoCoOp | 67.22 | 68.37 | 69.39 | 70.44 | 71.57 |
| | MaPLe | 66.60 | 71.60 | 75.30 | 79.47 | 83.57 |
| | PromptSRC | 69.40 | 73.40 | 77.13 | 80.97 | 83.83 |
| | DPD-LoRA (Ours) | 70.37 | 74.90 | 78.93 | 83.43 | 87.57 |
| Flowers102 | Linear probe CLIP | 69.74 | 85.07 | 92.02 | 96.10 | 97.37 |
| | CoOp | 77.53 | 87.33 | 92.17 | 94.97 | 97.07 |
| | CoCoOp | 72.08 | 75.79 | 78.40 | 84.30 | 87.84 |
| | MaPLe | 83.30 | 88.93 | 92.67 | 95.80 | 97.00 |
| | PromptSRC | 85.93 | 91.17 | 93.87 | 96.27 | 97.60 |
| | DPD-LoRA (Ours) | 85.77 | 91.03 | 94.37 | 96.30 | 98.00 |
| FGVCAircraft | Linear probe CLIP | 19.61 | 26.41 | 32.33 | 39.35 | 45.36 |
| | CoOp | 21.37 | 26.20 | 30.83 | 39.00 | 43.40 |
| | CoCoOp | 12.68 | 15.06 | 24.79 | 26.61 | 31.21 |
| | MaPLe | 26.73 | 30.90 | 34.87 | 42.00 | 48.40 |
| | PromptSRC | 27.67 | 31.70 | 37.47 | 43.27 | 50.83 |
| | DPD-LoRA (Ours) | 28.17 | 32.00 | 39.00 | 48.43 | 58.77 |
| SUN397 | Linear probe CLIP | 41.58 | 53.70 | 63.00 | 69.08 | 73.28 |
| | CoOp | 66.77 | 66.53 | 69.97 | 71.53 | 74.67 |
| | CoCoOp | 68.33 | 69.03 | 70.21 | 70.84 | 72.15 |
| | MaPLe | 64.77 | 67.10 | 70.67 | 73.23 | 75.53 |
| | PromptSRC | 69.67 | 71.60 | 74.00 | 75.73 | 77.23 |
| | DPD-LoRA (Ours) | 69.93 | 71.87 | 74.27 | 76.81 | 78.07 |
| OxfordPets | Linear probe CLIP | 44.06 | 58.37 | 71.17 | 78.36 | 85.34 |
| | CoOp | 90.37 | 89.80 | 92.57 | 91.27 | 91.87 |
| | CoCoOp | 91.27 | 92.64 | 92.81 | 93.45 | 93.34 |
| | MaPLe | 89.10 | 90.87 | 91.90 | 92.57 | 92.83 |
| | PromptSRC | 92.00 | 92.50 | 93.43 | 93.50 | 93.67 |
| | DPD-LoRA (Ours) | 91.87 | 92.71 | 93.80 | 93.83 | 94.02 |
| UCF101 | Linear probe CLIP | 53.66 | 65.78 | 73.28 | 79.34 | 82.11 |
| | CoOp | 71.23 | 73.43 | 77.10 | 80.20 | 82.23 |
| | CoCoOp | 70.30 | 73.51 | 74.82 | 77.14 | 78.14 |
| | MaPLe | 71.83 | 74.60 | 78.47 | 81.37 | 85.03 |
| | PromptSRC | 74.80 | 78.50 | 81.57 | 84.30 | 86.47 |
| | DPD-LoRA (Ours) | 74.13 | 78.57 | 81.5 | 85.13 | 86.82 |
| Food101 | Linear probe CLIP | 43.96 | 61.51 | 73.19 | 79.79 | 82.90 |
| | CoOp | 84.33 | 84.40 | 84.47 | 82.67 | 84.20 |
| | CoCoOp | 85.65 | 86.22 | 86.88 | 86.97 | 87.25 |
| | MaPLe | 80.50 | 81.47 | 81.77 | 83.60 | 85.33 |
| | PromptSRC | 84.87 | 85.70 | 86.17 | 86.90 | 87.50 |
| | DPD-LoRA (Ours) | 84.6 | 84.73 | 85.89 | 86.23 | 86.88 |
| Average | Linear probe CLIP | 45.83 | 57.98 | 68.01 | 74.47 | 78.79 |
| | CoOp | 67.56 | 70.65 | 74.02 | 76.98 | 79.89 |
| | CoCoOp | 66.79 | 67.65 | 71.21 | 72.96 | 74.90 |
| | MaPLe | 69.27 | 72.58 | 75.37 | 78.89 | 81.79 |
| | PromptSRC | 72.32 | 75.29 | 78.35 | 80.69 | 82.87 |
| | DPD-LoRA (Ours) | 72.62 | 75.60 | 78.73 | 81.46 | 84.22 |

