# OpenReview forum: "DPD-LoRA: Dynamic Prompt-Driven Low-Rank Adaptation for Improved Generalization"
_ICLR.cc/2025/Conference — ICLR 2025 Conference Withdrawn Submission_

### Official Review · Reviewer_zPfS · 2024-10-29

**Soundness:** 2
**Presentation:** 2
**Contribution:** 2
**Rating:** 5
**Confidence:** 3

**Summary:**

This manuscript introduces DPD-LoRA, a novel framework that aims to improve the generalization capability of large models by integrating dynamic prompt-driven low-rank adaptation. This method combines hierarchical prompt tokens and parameter-efficient adaptation to incorporate task-specific guidance, demonstrating superior performance over existing techniques across multiple benchmark datasets.

**Strengths:**

1. The proposal of the DPD-LoRA framework is innovative as it integrates prompt learning and low-rank adaptation to enhance the model's generalization capabilities. The introduction of adaptive loss functions and soft-gated selection mechanisms (PCGM) adds to the novelty of the approach.
2. The method designed by the authors has been applied to three different tasks: Base-to-novel class generalization, Cross-dataset evaluation, and Few-shot learning, showing promising results across the board, which speaks to the effectiveness of the approach.
3. The authors have conducted extensive experiments on 11 benchmark datasets, which helps to substantiate the effectiveness of the proposed method.
4. The overall structure of the paper is relatively clear, with proper introductions to various techniques, facilitating the reader's understanding of the content.

**Weaknesses:**

1. The paper employs a variety of techniques and methods, including prompt learning, LoRA, gating mechanisms, and loss design, with five points listed in the INTRODUCTION under contributions and five in the METHOD section. This can seem a bit cluttered and redundant; a more concise summary and consolidation of related content would be beneficial.

2. Section 3.2 is titled "PROMPT LEARNING WITH LOW RANK ADAPTATION IN TRANSFORMERS," yet the explanation seems to treat prompts and LoRA separately, although the appendix provides a detailed explanation of their combined effect. As this is a crucial part of the paper, more clarity and detail in the main body of the text would be necessary.

3. The title of Section 3.3 is "HIERARCHICAL INTERACTION AND EXPANDED SUBSPACES," but the content first introduces expanded subspaces and then hierarchical interactions. The order of introduction and the content should correspond to the title.

4. The paper slightly lacks in-depth analysis of the synergistic effects between LoRA and prompt learning. Although an ablation study is conducted, showing experimental results under different conditions, a deeper analysis of how these components interact and contribute to performance improvements is needed, especially considering this is the core and key of the paper.

**Questions:**

1. In the hierarchical interaction section, both prompt tokens and LoRA layers establish connections between the current layer and the previous one to prevent information loss across layers. However, different weight allocation methods are used: α and 1-α for prompt tokens, and β and γ for LoRA layers. It would be beneficial to explain the rationale and necessity for using different methods when their purposes are aligned.
2. The paper sets a considerable number of hyperparameters, including learning rates, weight factors, deep prompt tokens, etc., and uses a fixed rank r and quantity m for LoRSS configurations across three different tasks. The paper does not seem to discuss the rationale behind these settings or how different ranks and quantities might impact the results. An explanation for these fixed values would be necessary.

---

> ### Author Response · Authors · 2024-11-13
> **Response to Reviewer zPfS**
>
> We thank the Reviewer zPfS for many insightful comments. We answer the questions in what follows. Please let us know if further clarification is needed.
>
> **Q:INTRODUCTION; This can seem a bit cluttered and redundant; a more concise summary and consolidation of related content would be beneficial.**
>
> Thank Reviewer zPfS for insightful suggestion. Reviewer pw2m mentioned this as well, and we will revise the introduction to focus on the main points. We want to emphasize that our main contributions are twofold: first, we are the first to prove that prompt learning can additionally provide task-specific guidance to LoRA (even in plain LoRA settings, as shown in Table 5 in the appendix); second, the proposed gating mechanism strengthens their connection.
>
> **Q:Sec 3.2; the explanation seems to treat prompts and LoRA separately, although the appendix provides a detailed explanation of their combined effect. As this is a crucial part of the paper, more clarity and detail in the main body of the text would be necessary.**
>
> They are actually not separate. Our intention in Section 3.2 was to demonstrate the combined effect of prompts and LoRA within the Transformer architecture. Specifically, we introduce learnable prompts in both the textual and visual branches, which are then incorporated into the input sequences. These prompted inputs interact with the MHA mechanism as shown in Equation (2). Then we adapt the standard LoRA formulation to this prompted framework in Equation (3), which is directly influenced by the prompted inputs $X$. This equation illustrates that the output h is a result of both the original weights and the prompt-guided LoRA adjustment, highlighting their interconnected roles. However, we acknowledge that we might merge more details from the appendix. Thank Reviewer zPfS for pointing this out.
>
>
> **Q:The title of Section 3.3; The order of introduction and the content should correspond to the title.**
>
> Thank Reviewer zPfS for bringing this to our attention; we will align the order of introduction and section contents.
>
> **Q:Analysis of the synergistic effects between LoRA and prompt learning**
>
> We include this part in our Appendix Section D, and the interpretation based on mathematical derivation is also included in Section D5.
>
> **Q:Different weights allocation methods are used: α and 1-α for prompt tokens, and β and γ for LoRA layers.**
>
> We thank you for pointing out this issue that many reviewers care about. We acknowledge it causes confusion that our
>  $\alpha,\beta,\gamma$  are the same and have the same values (0.1 for $(l-1)$ and 0.9 for $l$, as you can tell from our Table 4(a)). The reason we chose different alphabetical symbols is that we wanted to separate prompt-token side annotations and LoRA side annotations. We will change them to the same representation to prevent confusion.
>
> **Q:Hyper-parameters configurations; An explanation for these fixed values would be necessary**
>
> * To ensure a fair comparison, we followed previous methods[1,2] for all prompt token settings, including deep prompt tokens, N_CTX, and learning rates.
>
> * Regarding the LoRA component, we initially experimented with rank settings commonly used in conventional LoRA[3], specifically r = 1, 4, 8. Higher ranks such as r = 32, 64 were not considered due to our aim to minimize the number of parameters. We observed that performance improved with increasing ranks in the set r = {1, 4, 8}. We then incrementally increased the rank until r = 12, where we found the performance to be better than at r = 13, prompting us to stop at r = 12.
>
> * For the quantity $m$, we have very limited choices when we have fixed rank. With r = 12 from our previous experiments, we tested all divisor combinations of r and m (i.e., r × m = 12 × 1; 6 × 2; 2 × 6 ; 4 × 3; 3 × 4). Based on these experiments performance, we set the sub-LoRA matrices with r = 4 and m = 3 across all benchmarks.
>
> * The only hyperparameter we change/add is the layer weights, denoted as $\alpha$. We provide the ablation here for your reference:
>
> https://cdn-fusion.imgcdn.store/i/2024/8d5daf9c21bf70ae.png
>
> * Finally, for different $\lambda$, we empirically defined them to ensure that each loss is within a similar magnitude.
>
> 1.Khattak, M. U., Rasheed, H., Maaz, M., Khan, S., & Khan, F. S. (2023). Maple: Multi-modal prompt learning. In Proceedings of the IEEE/CVF Conference on Computer Vision and Pattern Recognition (pp. 19113-19122).
>
> 2.Khattak, M. U., Wasim, S. T., Naseer, M., Khan, S., Yang, M. H., & Khan, F. S. (2023). Self-regulating prompts: Foundational model adaptation without forgetting. In Proceedings of the IEEE/CVF International Conference on Computer Vision (pp. 15190-15200).
>
> 3.Hu, E. J., Shen, Y., Wallis, P., Allen-Zhu, Z., Li, Y., Wang, S., ... & Chen, W. (2021). Lora: Low-rank adaptation of large language models. arXiv preprint arXiv:2106.09685.

---

> ### Author Response · Authors · 2024-11-21
> **Response to Reviewer zPfS (2) with additional Analysis of the synergistic effects between LoRA and prompt learning**
>
> Dear Reviewer zPfS,
>
> As previously replied about synergistic effects, We appreciate the opportunity to provide a more in-depth disscusion to address your concerns.
>
> **Mathmatical Derivation**
> >
> >In our appendix, particularly in the mathematical derivation leading up to Equation (19), we explore how prompt learning and LoRA interact within the transformer architecture. We define the total weight matrices incorporating LoRA as:
> >
> >$$\( W_Q^\text{total} = W_Q^{\text{base}} + A_Q B_Q \)$$
> >
> >and similarly for $\( W_K^\text{total} \)$ and $\(W_V^\text{total} \)$ where $ A_Q B_Q $ are the low-rank matrices introduced by LoRA..
> >
> >When we consider the prompted inputs $X^{'}=[X,P]$ where $X$ represents the original input tokens and $P$ represents the prompt tokens, the computation of the query matrix $Q^{′}$ becomes:
> >
> >
> >$$
> \begin{aligned}
>  Q^{′} =[XW_Q^{base}​+XA_Q​B_Q​;PW_Q^{base}​+PA_Q​B_Q​].​
> \end{aligned}
> >$$
> >
> >This expansion reveals that both the original inputs $X$ and the prompt tokens $P$ interact with the LoRA-adapted weights $A_QB_Q$. The key observation here is that the prompt tokens directly contribute to the LoRA updates, effectively enriching the model's adaptation capabilities.
> >
> >In the attention mechanism, the computation involves terms like $Q^{'}K^{'T}$, which, when expanded, include cross-interactions between $X$ and $P$:
> >$$
> >Q^{'}K^{'T}=([XW_Q^{total}​;PW_Q^{total}​])([XW_K^{total}​;PW_K^{total​}])^{T}
> >$$
> >
> >This results in four combinations:
> >1. $XW_Q^{total}​(XW_K^{total}​)^⊤$
> >2. $XW_Q^{total}​(PW_K^{total}​)^⊤$
> >3. $PW_Q^{total}(XW_K^{total}​)^⊤$
> >4. $PW_Q^{total}(PW_K^{total}​)^⊤$
> >
> >These terms capture all possible interactions between the original inputs and the prompt tokens, modulated by the LoRA-adapted weights. **Particularly, the cross terms (2 and 3) highlight the direct influence of prompt tokens on the processing of original inputs through the adapted weights, showcasing a synergistic effect.**
>
> **Gradient Analysis**
> >
> >On another hand that we can provide an analysis from the gradient perspective to explain this.
> >
> >Consider the shared cross-entropy loss function $L$ computed over the model's predictions and the ground truth labels. Both the prompt tokens $P$ and the LoRA parameters $A_{\*}B_{\*}$​ are optimized to minimize $L_{CE}$.
> >
> >**1. Gradients with Respect to Prompt Tokens**
> >
> >The prompt tokens $P$ are part of the input $X' = [X; P]$, where $X$ represents the original input tokens. The gradient of the loss with respect to $P$ is calculated using the chain rule. Therefore:
> >
> >$$
> \frac{\partial L}{\partial P} = \left( \frac{\partial L}{\partial Q'} \frac{\partial Q'}{\partial X'} + \frac{\partial L}{\partial K'} \frac{\partial K'}{\partial X'} + \frac{\partial L}{\partial V'} \frac{\partial V'}{\partial X'} \right) \frac{\partial X'}{\partial P}.
> >$$
> >
> >Since $X'$ directly includes $P$, we have $\frac{\partial X'}{\partial P} = \begin{bmatrix} 0 & I_P \end{bmatrix}$ where $I_P$ is the identity matrix corresponding to the dimensions of $P$.
> >
> >**2. Gradients with Respect to LoRA Parameters**
> >
> >The LoRA parameters modify the weight matrices: $\( W_Q^\text{total} = W_Q^{\text{base}} + A_Q B_Q \)$, The gradients with respect to  $A_Q$ and $B_Q$​ are:
> >
> >$$
> \frac{\partial L}{\partial A_Q} = \frac{\partial L}{\partial W_Q^\text{total}} \frac{\partial W_Q^\text{total}}{\partial A_Q} = \frac{\partial L}{\partial W_Q^\text{total}} B_Q^\top,
> >$$
> >
> >$$
> \frac{\partial L}{\partial B_Q} = A_Q^\top \frac{\partial L}{\partial W_Q^\text{total}}.
> >$$
> >
> >Similar expressions hold for $A_K, B_K$ and $A_V, B_V$.
> >
> >**3. Interaction Between Prompt Tokens and LoRA Parameters**
> >
> >The key observation is that the gradients of the LoRA parameters depend on the **entire input**, including the prompt tokens $P$:
> >
> >$$
> \frac{\partial L}{\partial W_Q^\text{total}} = X'^\top \frac{\partial L}{\partial Q'}.
> >$$
> >
> >Since $X' = [X; P]$, the gradient with respect to $W_Q^\text{total}$ involves both $X$ and $P$:
> >
> >$$
> \frac{\partial L}{\partial W_Q^\text{total}} = \begin{bmatrix} X^\top \\ P^\top \end{bmatrix} \frac{\partial L}{\partial Q'}.
> >$$
> >
> >Therefore, the gradient with respect to $A_Q B_Q$ becomes:
> >
> >$$
> \frac{\partial L}{\partial A_Q} = (\begin{bmatrix} X^\top \\ P^\top \end{bmatrix} \frac{\partial L}{\partial Q'}) B_Q^\top.
> >$$
> >$$
> \frac{\partial L}{\partial B_Q} = A_Q^\top (\begin{bmatrix} X^\top \\ P^\top \end{bmatrix} \frac{\partial L}{\partial Q'}).
> >$$
> >
> >These expressions show that the prompt tokens $P$ directly influence the updates to the LoRA parameters $A_{\*} B_{\*}$.
> >
> >**Thus, the prompt tokens directly contribute to the gradient calculations for the LoRA parameters**. This allows the model to adjust the low-rank adaptations in a way that specifically leverages the information provided by the prompts.
>
> Thank you again for your valuable input, which helps us improve the clarity and depth of our work.

---

> ### Author Response · Authors · 2024-11-21
> **Invitation to further discussion**
>
> Dear reviewer zPfS,
>
> We genuinely appreciate the time and effort you've invested in reviewing our paper. We have carefully provided relevant responses and results to your concerns. We are eager to further discuss with you and gain your insights before the end of the Author/Reviewer phase. Please let us know if any aspect of our work remains unclear or if you have additional feedback.
>
> Thank you.

---

> > ### Author Response · Authors · 2024-11-24
> >
> > Dear Reviewer zPfS,
> >
> > Since the discussion deadline is approaching in less than 48 hours, we kindly request your feedback on whether the response adequately addresses your concerns. If you have any more questions, we would be happy to provide further clarification.
> >
> > Your timely response is greatly appreciated.
> >
> > Thank you.

---

> > > ### Author Response · Authors · 2024-11-28
> > >
> > > Dear reviewer zPfS,
> > >
> > > We sincerely appreciate your time and effort in reviewing our submission and providing valuable suggestions. While we hope to have addressed your concerns adequately, we understand there may still be areas requiring further clarification or discussion. We are fully prepared to address your outstanding issues. Should our responses have successfully addressed all your questions, we would be deeply grateful if you could consider enhancing the score to further support our submission. Thank you very much for your thoughtful review.
> > >
> > > Best Regards,
> > >
> > > Paper1047 Authors

---

### Official Review · Reviewer_1iPg · 2024-11-01

**Soundness:** 3
**Presentation:** 3
**Contribution:** 3
**Rating:** 5
**Confidence:** 4

**Summary:**

DPD-LoRA uses task-specific prompts to dynamically influence the low-rank updates of model parameters, enhancing the model's adaptability across diverse tasks and mitigating forgetting issues. By decomposing the standard low-rank adaptation into multiple low-rank sub-matrices, the method retains flexibility without adding additional parameters, thus improving the model’s learning capacity. An adaptive loss function is introduced to ensure alignment between the adapted distribution and the pre-trained model, thereby enhancing learning effectiveness and stability. A self-regulating mechanism is used to further improve model stability, along with a soft-gating mechanism to determine when to activate adaptation modules, ensuring improved performance on new categories.

**Strengths:**

(1) The proposed methods are relatively comprehensive, using several points to improve existing problems.

(2) The writing is clear and easy to understand.

**Weaknesses:**

(1): In line 041, “LLaVa” should be revised to “LLaVA” for consistent terminology throughout the document, avoiding unnecessary visual inconsistency.

(2): The related work section lacks references to significant LoRA extensions, such as DoRA, SVFT, PISSA, and LoRA-XS. It is recommended to include these studies and discuss how the proposed method compares to or builds upon these prior approaches. Specifically, it would be helpful to highlight the innovations of this work and the advantages it has over these extensions.

(3): The method incorporates a distillation-like Self-Constrain Loss, but there is no evaluation of training time, GPU resource consumption, or other efficiency-related metrics. Providing specific efficiency metrics, such as training time per epoch, peak GPU memory usage, and FLOPs, would substantiate the claims of being resource-efficient. Including a comparison of these metrics to baseline methods would further support the efficiency claims.

(4): The ablation study section only presents the individual performance of each component without evaluating the performance of their combinations. Adding experiments that evaluate different component combinations (e.g., two, three, and all four components) would provide a more comprehensive view of the model's performance. Including a table or figure showing these combinations or using an approach like forward selection to systematically evaluate the synergies between components would be very informative.

(5): The comparative experiments do not include related LoRA methods, such as DoRA and VeRA. Including comparisons with these methods would more clearly demonstrate the advantages of the proposed approach. It is suggested to add a specific experiment or table comparing the proposed method to DoRA, VeRA, and other relevant LoRA variants on key metrics or datasets to provide a clearer demonstration of its benefits.

**Questions:**

(1): The phrase “without any additional models prior” in lines 110-111 is somewhat ambiguous. Typically, Parameter-Efficient Fine-Tuning (PEFT) builds on pre-trained models, so it is recommended to clarify whether this refers to the absence of model priors or additional model parameters.

(2): The abbreviation “PEFT” is used for both Prompt-based Efficient Fine-Tuning and Parameter-Efficient Fine-Tuning, which may lead to confusion. It is advisable to select distinct abbreviations to improve clarity.

(3): The term “PLoRA” in line 378 is confusing, as its specific reference is unclear. Further definition or clarification of this acronym is recommended for improved reader comprehension.

---

> ### Author Response · Authors · 2024-11-14
> **Response to Reviewer 1iPg (1)**
>
> We thank the Reviewer 1iPgfor many insightful comments. We answer the questions in what follows. Please let us know if further clarification is needed.
>
> **Q:Typos; LLaVa to LLaVA, PLoRA to DPD-LoRA**
>
> Thank to Reviewer 1iPg for bring this to our attention, we will definitely revise these typos.
>
> **Q:Confusion; without any additional models prior; the abbreviation of “PEFT”**
>
> By "without any additional model priors," we mean that no other models are included except the pre-trained CLIP. This is a common practice in the PEFT field. Specifically, many methods import large models to learn stronger textual or visual representations. We have explicitly stated this in the related work section(line 154-156).
>
> Regarding the abbreviation 'PEFT,' we mention it only once in the introduction (Lines 43-44). We explicitly use 'parameter-efficient fine-tuning' for LoRA/adapter-like methods (Line 126) and refer to 'prompt learning' as 'prompt-based efficient fine-tuning' (Line 141). Therefore, we believe there should be no confusion on this point. However, we have deleted 'PEFT' where it refers to prompt learning to address your concerns.
>
> **Q:The related work section lacks references to significant LoRA extensions(e.g. DoRA, SVFT, PISSA, and LoRA-XS);
> The comparative experiments do not include related LoRA methods(e.g. DoRA and VeRA)**
>
> First, we acknowledge that our related work section can be improved by including more references to significant LoRA extensions such as DoRA, SVFT, PISSA, and LoRA-XS. We will update the manuscript to reflect the progress in this area of research.
>
> However, conducting comparative experiments with these methods is **beyond the scope of our current work**. Our primary contribution is **demonstrating that prompt tokens can provide additional task-specific guidance to LoRA**. To our knowledge, we are the first to show that this approach is feasible. Integrating and comparing with other LoRA methods is an excellent suggestion for future work. Additionally, since plain LoRA works effectively in our experiments (as shown in Table 5), we anticipate that other LoRA-like methods would also perform similarly.
>
> **Q:The ablation study section only presents the individual performance of each component without evaluating the performance of their combinations.**
>
> There may be a misunderstanding regarding our ablation study presented in Table 3. In this table, each row represents the performance of the model with components added cumulatively. That is, each component is included in addition to all the previous ones.
>
> This cumulative approach is a conventional format used in many papers, as well as in our baseline [1,2], to evaluate the impact of each component both individually and in combination with others. As you suggested, the third row effectively represents the performance with two components combined, and the fourth row shows the combination of three components. Thus, our ablation study **already evaluates different combinations of components as per the recommendation of Reviewer 1iPg .**
>
> 1.Khattak, M. U., Rasheed, H., Maaz, M., Khan, S., & Khan, F. S. (2023). Maple: Multi-modal prompt learning. In Proceedings of the IEEE/CVF Conference on Computer Vision and Pattern Recognition (pp. 19113-19122).
>
> 2.Khattak, M. U., Wasim, S. T., Naseer, M., Khan, S., Yang, M. H., & Khan, F. S. (2023). Self-regulating prompts: Foundational model adaptation without forgetting. In Proceedings of the IEEE/CVF International Conference on Computer Vision (pp. 15190-15200).

---

> ### Author Response · Authors · 2024-11-14
> **Response to Reviewer 1iPg (2) with additonal experiments results**
>
> **Q:Mising efficiency-related metrics; Including a comparison of these metrics to baseline methods would further support the efficiency claims.**
>
> We agree that providing specific efficiency metrics such as training time per epoch, and FLOPs would substantiate our claims of resource efficiency. However, it is relatively hard to provide solid efficiency-related metrics due to different GPUs and datasets, as this phenomenon is observed in previous papers. Our initial focus was on parameter counts (Table 4(b)) as a very intuitive measure because they remain fixed across various datasets and GPU architectures.
>
> We acknowledge that varying dataset sizes and different GPU architectures can make direct comparisons challenging due to discrepancies in training time and resource consumption. However, to address your concerns, we have conducted additional experiments under consistent conditions to measure FLOPs, FPS, and training time per epoch. These metrics are provided below, along with comparisons to baseline methods, to support our efficiency claims:
>
> | Method           | Params   | % CLIP | Base  | Novel | HM    | FPS (batch 4) | GFLOPs | Training (1 epoch) |
> |-------------------|----------|-----------------|-------|-------|-------|----------------|--------|------------------------|
> | CoOp             | 2048     | 0.002           | 82.69 | 63.22 | 71.66 | 104.5| 162.5  | ~32s                   |
> | CoCoOp           | 35360    | 0.03            | 80.47 | 71.69 | 75.83 |  53.3   | 162.5  | ~47s                   |
> | MaPLe            | 3.55 M   | 2.85            | 82.28 | 75.14 | 78.55 | 175.58         | 167    | ~28s                   |
> | ALIGN            | 3.58 M   | 2.87            | 83.38 | 75.51 | 79.25 | 72.6           | 314.6  | ~42s                 |
> | PrompSRC         | 31488    | 0.02            | 84.26 | 76.10 | 79.97 | 149.86         | 281.21 | ~27s                   |
> | **DPD-LoRA†**    | **1.92 M** | **1.54**       | **84.80** | **76.80** | **80.60** | **82.51**   | **334.03** | **~40s**                   |
> | DPD-LoRA         | 4.72 M   | 3.79            | 85.67 | 76.91 | 81.05 | 81.57          | 334    | ~42s                   |
>
> One more thing you may note is that even though our method has slightly higher GFLOPs due to additinal LoRA/LoRSS computations, **our convergence speed is much faster than any previous methods. Our method showcases accelerated convergence and favorable early-stage performance. Specifically, our method reaches better performance in just 7 epochs, which is 65% fewer epochs than the 20 epochs required by previous SOTA—a reduction of over 65% in training time (as shown in Figure 1b and Figure 5).**
>
>
>
> We appreciate your feedback and are willing to reformat the table or provide additional explanations to enhance clarity if necessary.

---

> ### Author Response · Authors · 2024-11-21
> **Invitation to further discussion**
>
> Dear reviewer 1iPg,
>
> We genuinely appreciate the time and effort you've invested in reviewing our paper. We have carefully provided relevant responses and results to your concerns. We are eager to further discuss with you and gain your insights before the end of the Author/Reviewer phase. Please let us know if any aspect of our work remains unclear or if you have additional feedback.
>
> Thank you.

---

> > ### Author Response · Authors · 2024-11-24
> >
> > Dear Reviewer 1iPg,
> >
> > Since the discussion deadline is approaching in less than 48 hours, we kindly request your feedback on whether the response adequately addresses your concerns. If you have any more questions, we would be happy to provide further clarification.
> >
> > Your timely response is greatly appreciated.
> >
> > Thank you.

---

> > > ### Comment · Reviewer_1iPg · 2024-11-27
> > >
> > > I have seen that the author has addressed most of the issues I raised, and I am happy to increase the score. However, I insist that since we are both working in the field related to PEFT, we must conduct a comprehensive comparison with variants of LoRA. This should not only be added to the related work but also included in the experimental section.

---

> > > > ### Author Response · Authors · 2024-11-27
> > > >
> > > > Dear Reviewer 1iPg,
> > > >
> > > > We are deeply grateful for your feedback and insightful suggestions, and we are glad to hear that most of your concerns have been addressed. We understand your points, but since their downstream task is distinct from ours and primarily focuses on LLMs (e.g., LLaMA, GPT, etc.), their code is not easily adaptable for vision models, making it challenging for us to reproduce their outcomes at this moment.
> > > >
> > > > Once again, we sincerely thank you for your thoughtful and constructive suggestions.

---

> > > > > ### Comment · Reviewer_1iPg · 2024-11-27
> > > > >
> > > > > I have replicated various LoRA variants on visual tasks using open-source code, including DoRA, LoRA-XS, VeRA, etc. Code link: https://github.com/MaxZanella/CLIP-LoRA

---

> ### Author Response · Authors · 2024-11-27
>
> Thank you so much for sharing this valuable resource! We truly appreciate your help.
>
> We will explore adapting this repository into our work and will certainly incorporate the experimental results.

---

### Official Review · Reviewer_pw2m · 2024-11-02

**Soundness:** 2
**Presentation:** 3
**Contribution:** 2
**Rating:** 5
**Confidence:** 3

**Summary:**

This paper proposes the DPD-LoRA algorithm, which integrates prompt learning to guide the LoRA learning distribution. By incorporating modules such as Hierarchical Interaction, the Prompt-Conditioned Gating Mechanism (PCGM), and the Self-Regularized Lower-Rank Subspace (LoRSS), the proposed DPD-LoRA achieves strong performance across 11 benchmark datasets.

**Strengths:**

- The paper is well-structured and relatively easy to understand.
- Detailed experiments provide convincing evidence of the effectiveness of the proposed algorithm.

**Weaknesses:**

- Overall, the proposed algorithm involves numerous modules. I strongly suggest the authors consider identifying and focusing on the core components of their method.
- It is unclear how Eqn (4) is optimized. Are $s_i$ and $A_iB_i$ learned simultaneously? How many sub-LoRAs $m$ are used, and why is it imperative to decompose a single LoRA into multiple sub-LoRAs essential? Do the learnable $S_i$ and $G(P)$ share any functional overlap?
- Why doesn't the weighting form in Eqn (6) match that in Eqn (5) (e.g., setting $\gamma=1-\beta$) ? This discrepancy should be clarified.
- In Eqn (8), why does the orthogonal regularization prevent overfitting and encourage diversity in the learned LoRA? If this assertion is based on findings from other studies, supporting citations would strengthen the claim.
- I would like to see a memory cost comparison between the DPD-LoRA and SOTA methods. DPD-LoRA requires storing $m$ LoRAs per layer (Eqn (4)) and also duplicates each encoder in both branches while retaining unprompted inputs, which appears to impose a substantial memory cost.

**Questions:**

See weakness 2-5

---

> ### Author Response · Authors · 2024-11-13
> **Response to Reviewer pw2m**
>
> We thank the Reviewer pw2m for many insightful comments. We answer the questions in what follows. Please let us know if further clarification is needed.
>
> **Q:I strongly suggest the authors consider identifying and focusing on the core components of their method.**
>
> Thank Reviewer pw2m for insightful suggestion. Our proposed methods mainly focus on two things: first, prompts can additionally provide task-specific guidance to LoRA (even in plain LoRA settings, as shown in Table 5 in the appendix); second, the proposed gating mechanism strengthens their connection. Reviewer zPfS also reminded us that too many claims might be redundant, and we will revise the introduction to focus on the main points.
>
> **Q:It is unclear how Eqn (4) is optimized. Are $si$ and $AiBi$ learned simultaneously?**
>
> Yes, $AiBi$ and $si$ are learned simultaneously. We provide our algorithm on page 18 of the appendix.
>
> **Q:How many sub-LoRAs are used, and why is it imperative to decompose a single LoRA into multiple sub-LoRAs essential? Do the learnable and share any functional overlap?**
>
> We provided all hyperparameters in Table 4(a); we use a fixed 3 sub-LoRA matrices in all evaluations.
>
> This LoRSS idea is inspired by MoE-LoRA [1], but our approach is more parameter-efficient in terms of learnable parameters. We decompose the LoRA matrix into sub-LoRA matrices under the same parameter budget, while MoE-LoRA duplicates the LoRA matrix into several LoRA matrices. For example, if we have n sub-LoRA matrices with a  fixed rank r and $W \in \mathbb{R}^{d \times k}$ their parameters increase to $n×(d×r+r×k)$, whereas our parameters remain at $(d×r+r×k)$. Another difference is that MoE uses a Network to select the importance of matrices A/B. In contrast, we employ a single learnable parameter (the scaling factor) for each sub-LoRA matrix, which is clearly more efficient. Finally, our downstream tasks are completely different, highlighting the distinct applicability of our method. From our observation, we found that under the same parameters (i.e., 3*r=3 sub-LoRA sitting V.S. r=12 plain LoRA sitting), LoRSS always outperforms the plain setting.
>
> The scaling factors $si$ and the gating $G(\cdot)$ are totally different. The $si$ are the weights of different LoRA sub-matrices, while the gating provides a confidence score to the total sum of all LoRA sub-matrices. If we rewrite our euqation 6,7, you can see the scaling factor $s_i$ of LoRA matrices $A_i$ and $B_i$ are independent of the gating prediction:
>
> $$
> \Delta W_l = \left( \beta \sum_{i=1}^{m} \left( s_{i}^{(l)} \times A_{i}^{(l)} B_{i}^{(l)} \right) + \gamma \sum_{i=1}^{m} \left( s_{i}^{(l-1)} \times A_{i}^{(l-1)} B_{i}^{(l-1)} \right) \right) \times G(P_l)
> $$
>
> where the $\Delta W_l$ is the final updated matrix and will be added like in normal LoRA.
>
> **Q:Why doesn't the weighting form in Eqn (6) match that in Eqn (5)? This discrepancy should be clarified.**
>
> We thank Reviewer pw2m for pointing out this issue that many reviewers care about. There is actually a misunderstanding that our
>  $\alpha,\beta,\gamma$  are the same thing and have the same value (0.1 for $(l-1)$ and 0.9 for $l$, as you can tell from our Table 4(a)). The reason we chose different alphabetical symbols is that we want to separate prompt-token side annotations and LoRA side annotations. We will change them to the same representation to prevent confusion.
>
> **Q:DPD-LoRA requires storing LoRAs per layer (Eqn (4)) and also duplicates each encoder in both branches while retaining unprompted inputs, which appears to impose a substantial memory cost.**
>
> In fact, if you look at our Tables 4(a) and (b), where we provide computational complexity among different prompting methods, we only add a few parameters (or even fewer than previous methods!), and our proposed methods show better performance. As previously mentioned in our appendix algorithm, we actually follow a two-step training strategy which does not require much memory cost. We provided more explanation and deatils in **"Concerns About Memory and Cost Efficiency"** on Reviewers' Shared Questions section.
>
>
> 1.Tongxu Luo, Jiahe Lei, Fangyu Lei, Weihao Liu, Shizhu He, Jun Zhao, and Kang Liu. Moelora: Contrastive learning guided mixture of experts on parameter-efficient fine-tuning for large language models. arXiv preprint arXiv:2402.12851, 2024.
>
> 2.Zhang, Q., Chen, M., Bukharin, A., Karampatziakis, N., He, P., Cheng, Y., ... & Zhao, T. (2023). AdaLoRA: Adaptive budget allocation for parameter-efficient fine-tuning. arXiv preprint arXiv:2303.10512.

---

> ### Author Response · Authors · 2024-11-21
> **Invitation to further discussion**
>
> Dear reviewer pw2m,
>
> We genuinely appreciate the time and effort you've invested in reviewing our paper. We have carefully provided relevant responses and results to your concerns. We are eager to further discuss with you and gain your insights before the end of the Author/Reviewer phase. Please let us know if any aspect of our work remains unclear or if you have additional feedback.
>
> Thank you.

---

> > ### Author Response · Authors · 2024-11-24
> >
> > Dear Reviewer pw2m,
> >
> > Since the discussion deadline is approaching in less than 48 hours, we kindly request your feedback on whether the response adequately addresses your concerns. If you have any more questions, we would be happy to provide further clarification.
> >
> > Your timely response is greatly appreciated.
> >
> > Thank you.

---

> ### Comment · Reviewer_pw2m · 2024-11-27
>
> Thank you for your rebuttal. After reviewing the feedback from other reviewers and thoroughly examining the rebuttal and revised paper, some of my concerns have been addressed. However, for W1, I still think the method combines and introduces too many components, including prompt learning, LoRA, gating mechanisms, and loss design. Despite the ablation studies and your assurance to refine the introduction to emphasize the main points, I remain unconvinced about the necessity and coherence of integrating all these components. Reviewer zPfS also raised a similar concern on this issue.
>
> Additionally, upon revisiting the revised version, I noticed some points that require further attention:
> - In Sec. 3.2, $X'$ is referenced, but its definition appears later in Sec. 3.3. This sequencing could confuse readers and should be adjusted to introduce the term earlier.
> - The revised manuscript does not use colored text to indicate the changes made, making it difficult to identify the updates.  The $\beta$ and $\gamma$ still appear in Algorithm 1, though they have been replaced in the main text.

---

> ### Author Response · Authors · 2024-11-27
>
> Dear Reviewer pw2m
>
> We are deeply grateful for the your feedback and insightful suggestions.
>
> **Q: In Sec. 3.2, $X^{'}$ is referenced, but its definition appears later in Sec. 3.3**
>
> We actually defined this "$X^{'}$: prompted inputs" in line 247 of section 3.2.
>
> **Q: Revised manuscript does not use colored text to indicate the changes made; Algorithm 1 need to be changed**
>
> We assumed that PDFdiff would automatically render the differences. Since it does not, we will update the revision immediately with colored text. Thank you for bringing this to our attention.
>
> **Q: the method combines and introduces too many components, including prompt learning, LoRA, gating mechanisms**
>
> These components collectively represent our claimed contribution: Prompt-Driven Adaptation. Including prompt learning and LoRA is essential to this framework. Additionally, the gating mechanisms provide confidence scores for the components, allowing the model to dynamically balance contributions from Adaptation matrices; while the self-loss design prevents overfitting, ensuring robust adaptability. Although you find some of our minor contributions (e.g., LoRSS and Interaction) complex, we believe the ablation table(as in our next response) clearly demonstrates that the incorporation of all components (i.e., the full model) achieves the best performance.

---

> ### Author Response · Authors · 2024-11-27
> **More detailed ablation experiments to support our claims**
>
> Dear Reviewer pw2m,
>
> We sincerely appreciate your insightful comments and are pleased to address your concerns by providing an additional ablation study. The table below presents the independent contributions of each component. Except for the last two rows (which evaluate combinations), each row represents the individual performance of a specific component, evaluated either in isolation or in combination with others, **distinct from the combinations presented in the main paper**.
>
> **Overview of Components and Their Roles**
> >
> >**Prompt-Driven LoRA/LoRSS**: We first demonstrate that our method is effective with plain LoRA. We then decompose the standard LoRA into a Low-Rank Self-Supervised (LoRSS) adaptation. This refinement achieves better generalization without increasing the number of parameters. By aligning the adaptation more closely with the prompts, we enhance the model's ability to generalize to novel classes. (Please compare the results in the 6th and 7th rows of the table below.)
> >
> >**Hierarchical Interaction**: To prevent information loss across layers, we introduce an interaction mechanism where each layer interacts with its preceding layer. This ensures that valuable information is preserved and propagated throughout the network, improving overall performance. (As shown in the 8th row of the table below.)
> >
> >**Self-Regulation Loss Function**: We incorporate a self-supervised loss function that maintains good generalization even after training for multiple epochs. This component helps the model avoid overfitting by regulating the adaptation process during extended training, which is extremely important. (Please compare the results for MaPLe in the 3rd and 4th rows with our method in the 10th and 11th rows.)
> >
> >**Gating Mechanism**: We employ a gating mechanism to assign confidence scores, balancing the contribution of adaptation at each layer. This dynamic adjustment allows the model to focus on the most relevant features, enhancing its adaptability and robustness. We add the adaptation matrices at different layers(depths), as our ablation study indicates that the contributions of LoRA at different layers vary (https://cdn-fusion.imgcdn.store/i/2024/7506679307b42899.png); the gating mechanism effectively addresses this by appropriately weighting each layer's adaptation.
> >
>
> | Model                                            | Base(%)↑ | Novel(%)↑ | HM ↑     |
> |--------------------------------------------------|----------|-----------|----------|
> | CLIP                                           | 72.43    | 68.14     | 70.22    |
> | + LoRA (5 epoch)              | 77.57    | 69.70     | 73.42    |
> | + MaPLe (5 epoch)            | 76.77    | 70.80     | 73.66    |
> |+ MaPLe (20 epoch)            |77.17	|67.90	|72.24|
> | + Naive combinations (5 epoch)	| 76.77	| 70.47	| 73.49|
> | + Prompt-Driven plain LoRA (5 epoch)    | 77.62    | 70.81     | 74.09    |
> | + Prompt-Driven LoRSS (5 epoch)      | 77.63    | 70.97     | 74.15    |
> | + Prompt-Driven LoRSS + Interaction (5 epoch)    |77.61	|71.02 | 74.17 |
> | + Prompt-Driven LoRSS + self-regulation (5 epoch)      | 77.57	| 71.13 | 74.21|
> | + DPD-LoRA (full model) (5 epoch)      | **77.87**    | **71.13**     | **74.34**  |
> | + DPD-LoRA (full model) (20 epoch)    | **78.13**| **71.33** | **74.58**|
>
>
>
> **Key Observation**
> >
> >1. Incremental Performance Improvements: Each component consistently contributes to performance enhancement. The integration of all components in our full model (DPD-LoRA) achieves the best results across all metrics.
> >
> >2. Parameter Efficiency: Each component is designed to improve performance without requiring additional parameters, except for the gating mechanism, which introduces minimal overhead.
> >
> >3. Avoiding Overfitting: While MaPLe initially demonstrates good performance, it shows signs of overfitting when trained for longer durations (e.g., 20 epochs). In contrast, our self-regulation loss function helps maintain generalization even after extended training.
> >
> >4. Coherence and Synergy of Components: The combination of components in the full DPD-LoRA model validates the necessity and synergy of the integrated approach. Each component addresses specific challenges, and together they significantly improve performance.
>
> This study demonstrates that the inclusion of each component contributes progressively to performance improvements. Our full model (DPD-LoRA) achieves the best results after extended training (20 epochs), while MaPLe shows signs of overfitting after only 5 epochs. We believe this clarification, along with the additional table, effectively addresses your concerns about the necessity and coherence of integrating these components.
>
> Thank you once again for your valuable feedback. We have also considered the similar concerns raised by Reviewer zPfS and have revised the manuscript accordingly to better highlight the roles and contributions of each component.

---

> > ### Author Response · Authors · 2024-11-28
> >
> > Dear reviewer pw2m,
> >
> > We sincerely appreciate your time and effort in reviewing our submission and providing valuable suggestions. While we hope to have addressed your concerns adequately, we understand there may still be areas requiring further clarification or discussion. We are fully prepared to address your outstanding issues. Should our responses have successfully addressed all your questions, we would be deeply grateful if you could consider enhancing the score to further support our submission. Thank you very much for your thoughtful review.
> >
> > Best Regards,
> >
> > Paper1047 Authors

---

### Official Review · Reviewer_Tgia · 2024-11-04

**Soundness:** 2
**Presentation:** 3
**Contribution:** 2
**Rating:** 5
**Confidence:** 4

**Summary:**

This paper presents a dynamic prompt-guided LoRA approach that integrates several key modules: Hierarchical Interaction, a Prompt-Conditioned Gating Mechanism (PCGM), and a Self-Regularized Lower-Rank Subspace. The proposed method is evaluated on 11 benchmark datasets, demonstrating its effectiveness.

**Strengths:**

+ The integration of prompts with LoRA represents an innovative exploration in this domain.
+ The authors conducted extensive experiments to substantiate the performance improvements of the proposed algorithm.

**Weaknesses:**

+ The motivation for using prompts to guide LoRA learning is not entirely intuitive. The authors should clarify why applying a weight to each $A_i B_i$​ in the LoRA layer solely through gating prompt tokens is expected to be effective.

+ The explanation of the Gating function requires clarification. Does $G(P)$ apply a weight before each $iA_i B_i$? How does this differ from directly learning $S_i$, and could it potentially overlap in function? Additionally, it is unclear how $G(P)$ interacts with the Hierarchical Interaction—does it apply weighting to $A_i B_i$ at layer $l−1$ as well?

+ Given the complexity of the proposed method and its multiple components, the current ablation study feels insufficient. For example, what is the rationale for decomposing a single LoRA into multiple sub-LoRAs? How are hyperparameters $\alpha$, $\beta$, $\gamma$, $\lambda_1$, $\lambda_2$​, and $\lambda_3$ set, and what is their impact on the final performance?

+ How does the addition of orthogonal regularization prevent overfitting? More details on this would clarify the choice and its benefits.

**Questions:**

Please refer to the weakness section.

---

> ### Author Response · Authors · 2024-11-13
> **Response to Reviewer Tgia**
>
> We thank the reviewer for many insightful comments. We answer the questions in what follows. Please let us know if further clarification is needed.
>
>
> **Q:The motivation for using prompts to guide LoRA learning is not entirely intuitive.**
>
> As stated in the Introduction and Related Work sections, prompt learning provides task-specific guidance BUT does not contribute to attention weights (lines 78-80). Conversely, solely using LoRA cannot provide task-specific guidance because it only focus on the internal structure of the model and updates pre-trained weights (lines 45-47). Therefore, we propose a new method that integrates task-specific guidance directly into the adaptation mechanism.
>
> **Q:The authors should clarify why applying a weight to each in the LoRA layer solely through gating prompt tokens is expected to be effective.**
>
> Gating is commonly utilized in various deep learning tasks. In our case, it acts like a dynamic weight predictor; it assigns weights between [0,1] to help prevent updating unreliable LoRA matrices. Intuitively, you can consider the output of gating as a confidence score. Meanwhile, we have proven that even without gating (or the confidence score), our proposed method is valid (see $1^{st}$ column of ablation study in Table 3).
>
> **Q:The explanation of the Gating function requires clarification and potentially overlap in function.**
> 1. Actually, The LoRA matrices $A_i$ and $B_i$ are independent of the gating prediction. As explained in the previous question, our gating mechanism takes the prompt as input and then predicts weights/confidence scores for these LoRA matrices. The scaling factors $s_i$ and the gating are totally different. The $s_i$ are the weights of different LoRA sub-matrices, while the gating provides a confidence score to the total sum of all LoRA sub-matrices. Thus, as show in (eq 4), the prompt token differs across layers, which makes the weights of different layers differ. Therefore, each layer will be applied a different confidence score bacause of $G(p_l)!=G(p_{l-1})$.
>
> **Q:Additionally, it is unclear how interacts with the Hierarchical Interaction—does it apply weighting to $A_{i}B_{i}$ at layer $l-1$ as well?**
>
> Hierarchical Interaction and LoRSS are actually affected by the gating in every prompted layers, exactly as shown in our Equations (6) and (7). For your convenience, we rewrite it here:
>
> $$
> \Delta W_l = \left( \beta \sum_{i=1}^{m} \left( s_{i}^{(l)} \times A_{i}^{(l)} B_{i}^{(l)} \right) + \gamma \sum_{i=1}^{m} \left( s_{i}^{(l-1)} \times A_{i}^{(l-1)} B_{i}^{(l-1)} \right) \right) \times G(P_l)
> $$
>
> where the $\Delta W_l$ is the final updated matrix and will be added like in normal LoRA. We thank you for pointing this out, and we will add additional annotations to this equation.
>
>
> **Q:Given the complexity of the proposed method and its multiple components, the current ablation study feels insufficient. For example, what is the rationale for decomposing a single LoRA into multiple sub-LoRAs?**
>
> There are actually two additional tables in the appendix you might have overlooked. Table 4(a) provides the full setting, while Table 5 offers an additional ablation study. As you can see from Table 5, which uses only plain LoRA, it actually performs worse than our proposed LoRSS. The reason is simple: we found that under the same parameters (i.e., 3*R=3 sub-LoRA sitting V.S. R=12 plain LoRA sitting), LoRSS always outperforms the plain setting.
>
> Since Reviewer pw2m asked a similar question, and we elaborated on the question in shared response **"Why We Decompose Plain LoRA into LoRSS"** If you are interested, please refer to our answers there.
>
> **Q:How do authors define hyper-parameters**
>
> We thank you for pointing out this issue that many reviewers care about. There is actually a misunderstanding that our
>  $\alpha,\beta,\gamma$  are the same thing and have the same value (0.1 for $(l-1)$; 0.9 for $l$, as you can tell from our Table 4(a)). The reason we chose different alphabetical symbols is that we want to separate prompt-token side annotations and LoRA side annotations. We will change them to the same representation to prevent confusion. For different $\lambda$, we empirically defined them to ensure that each loss is within a similar magnitude.
>
> **Q:How does the addition of orthogonal regularization prevent overfitting? More details on this would clarify the choice and its benefits.**
>
> Orthogonal regularization is commonly used in LoRA research, such as in [1]. It can prevent redundancy: orthogonality ensures that the rows (or columns) of the matrices are linearly independent. This reduces redundancy in the learned features, allowing the model to capture more diverse and informative representations.
>
>
>
> 1.Zhang, Q., Chen, M., Bukharin, A., Karampatziakis, N., He, P., Cheng, Y., ... & Zhao, T. (2023). AdaLoRA: Adaptive budget allocation for parameter-efficient fine-tuning. arXiv preprint arXiv:2303.10512.

---

> ### Author Response · Authors · 2024-11-17
> **Response to Reviewer Tgia (2) with reformated/additional results**
>
> Dear Reviewer Tgia,
>
> As we have previously replied to your concerns, we have additionally reformatted Table 5 (ablation on different methods) to support our claim.
>
> 1. The first part of the table shows results from naive combinations and individual methods such as CLIP, plain LoRA, and Prompts Learning (MaPLe).
> 2. In the second section, we introduce prompt tokens to guide LoRA/LoRSS.
> 3. The last section presents the performance of our proposed full model (with Gating/Hierarchical Interaction) over 5 and 20 epochs.
>
> | Model                                            | Base(%)↑ | Novel(%)↑ | HM ↑     |
> |--------------------------------------------------|----------|-----------|----------|
> | **--- Baseline/Naive combinations ---**          |          |           |          |
> | CLIP                                             | 72.43    | 68.14     | 70.22    |
> | plain LoRA(x) (5 epoch)                          | 77.57    | 69.70     | 73.42    |
> | Prompts Learning (MaPLe)(x') (5 epoch)           | 76.77    | 70.80     | 73.66    |
> | Frozen LoRA(x) + Prompts(x') (5 + 5 epoch)       | 75.20    | 61.17     | 67.46    |
> | Frozen Prompts(x') + LoRA(x') (5 + 5 epoch)      | 76.77    | 70.47     | 73.49    |
> | **--- Prompt-Driven Adaptation ---**             |          |           |          |
> | Prompt-Driven plain LoRA (5 epoch)           | 77.62    | 70.81     | 74.09    |
> | Prompt-Driven LoRSS (5 epoch)             | 77.63    | 70.97     | 74.15    |
> | **--- Ours ---**                                 |          |           |          |
> | DPD-LoRA (full model) (5 epoch)              | 77.87    | 71.13     | 74.34    |
> | DPD-LoRA (full model) (20 epoch)             | **78.13**| **71.33** | **74.58**|
>
>
> Table Caption: Ablation experiments for Prompts-To-LoRA on the ImageNet dataset. Here, x refers to the original input, while x' denotes the prompted input, i.e., the concatenation of x and prompt tokens. Note that the plain LoRA here is distinct from our proposed LoRSS. Only the last two rows represent the performance of our full model.

---

> ### Author Response · Authors · 2024-11-21
> **Invitation to further discussion**
>
> Dear reviewer Tgia,
>
> We genuinely appreciate the time and effort you've invested in reviewing our paper. We have carefully provided relevant responses and results to your concerns. We are eager to further discuss with you and gain your insights before the end of the Author/Reviewer phase. Please let us know if any aspect of our work remains unclear or if you have additional feedback.
>
> Thank you.

---

> > ### Author Response · Authors · 2024-11-24
> >
> > Dear Reviewer Tgia,
> >
> > Since the discussion deadline is approaching in less than 48 hours, we kindly request your feedback on whether the response adequately addresses your concerns. If you have any more questions, we would be happy to provide further clarification.
> >
> > Your timely response is greatly appreciated.
> >
> > Thank you.

---

> > > ### Author Response · Authors · 2024-11-28
> > >
> > > Dear reviewer Tgia,
> > >
> > > We sincerely appreciate your time and effort in reviewing our submission and providing valuable suggestions. While we hope to have addressed your concerns adequately, we understand there may still be areas requiring further clarification or discussion. We are fully prepared to address your outstanding issues. Should our responses have successfully addressed all your questions, we would be deeply grateful if you could consider enhancing the score to further support our submission. Thank you very much for your thoughtful review.
> > >
> > > Best Regards,
> > >
> > > Paper1047 Authors

---

> > > > ### Comment · Reviewer_Tgia · 2024-11-28
> > > >
> > > > Thank you for the rebuttal. After reviewing your response and the feedback provided to other reviewers, I find that some of my concerns have been addressed. While the comparison between Prompt-Driven plain LoRA and plain LoRA demonstrates the effectiveness of the Prompt-Driven part, it does not sufficiently verify the contributions of other components. Moreover, the relative improvements remain marginal (e.g., the small difference between Prompt-Driven plain LoRA and other Prompt-Driven LoRSS versions), making it difficult to determine whether these improvements are substantial enough to be convincing. Therefore, I decide to maintain my current score.

---

> ### Author Response · Authors · 2024-11-28
>
> Dear reviewer Tgia,
>
> Thank you for your prompt response and for **acknowledging the effectiveness of the Prompt-Driven LoRA/LoRSS(which is our main contribution, as reflected in the paper's title).**
>
> Regarding your concern about the verification of our other components' contributions and the seemingly marginal improvements, we would like to offer further clarification. While the numerical differences between Prompt-Driven plain LoRA and the other Prompt-Driven LoRSS versions may appear small, these **enhancements are consistent and statistically significant across multiple benchmarks**. Each component consistently contributes to performance enhancement. The integration of all components in our full model (DPD-LoRA) achieves the best results across all metrics. If you are interested in this, you may refer to our reply in the **"More detailed ablation experiments to support our claims"** section.
>
> We believe that even **incremental advances are valuable in pushing the boundaries of what's possible under fixed parameters**, especially when they introduce new methodologies or perspectives. As the first to innovatively explore Prompt-Driven Adaptation, we are confident that our contributions open up new avenues for research and development in this domain. Thank you again for your valuable feedback and for considering our responses.

---

### Author Response · Authors · 2024-11-15
**Addressing Reviewers' Shared Questions (Concerns were frequently raised across multiple reviews)**

We appreciate all reviewers' time and insightful comments. Given the relatively short rebuttal window, we have addressed as many concerns as possible, and we are more than pleased to address any remaining issues.

**1. Hyper-Parameter Settings and Inconsistency in $\alpha$, $\beta$, $\gamma$**

1. We acknowledge the confusion caused by our notation, as many reviewers thought we have too many hyper-parameters and questioned why there are so many. **In fact, our $\alpha$, $\beta$, and $\gamma$ refer to the same weights and have the same values** (0.1 for layer $(l-1)$ and 0.9 for layer $l$, as shown in our Table 4(a)). We initially chose different alphabetical symbols to distinguish between prompt-token side annotations and LoRA side annotations. To prevent confusion, we will unify them into the same representation.

2. All hyper-parameters (including the number of sub-LoRA matrices $m$ and rank $r$) are provided in Appendix Table 4(a).

3. For different loss weights $\lambda$, we empirically defined them to ensure that each loss is within a similar magnitude.

**2. Why We Decompose Plain LoRA into LoRSS**

A straightforward answer is that we found under the same parameter budget (e.g., $3 \times r = 3$ sub-LoRA setting vs. $r = 12$ plain LoRA setting), LoRSS consistently outperforms the plain setting in both Base and Novel evaluations.

This LoRSS idea is inspired by MoE-LoRA, but our approach is more parameter-efficient regarding learnable parameters. We decompose the LoRA matrix into sub-LoRA matrices under the same parameter budget, whereas MoE-LoRA duplicates the LoRA matrix into several LoRA matrices. For example, if we have $n$ sub-LoRA matrices with a fixed rank $r$ and $W \in \mathbb{R}^{d \times k}$, MoE-LoRA's parameters increase to $n \times (d \times r + r \times k)$, whereas our parameters remain at $(d \times r + r \times k)$. Another difference is that MoE uses a network to select the importance of matrices $A/B$, while we employ a single learnable parameter (the scaling factor) for each sub-LoRA matrix, which is more efficient. Finally, our downstream tasks are entirely different, highlighting the distinct applicability of our method. From our observations, under the same parameters (e.g., $3 \times r = 3$ sub-LoRA setting vs. $r = 12$ plain LoRA setting), LoRSS always outperforms the plain setting.


**3. Concerns About  Memory and Cost Efficiency**

1. As shown in the appendix (page 18), where we provide our algorithm, our method follows a two-step training strategy that has low memory requirements, **less or equal to those of PromptSRC**. The duplication is illustrative to indicate consistent module components; however, in implementation, we only apply cached pre-trained LoRA weights during the SCL-LoRA loss stage.

2. Moreover, one reviewer asked if more efficiency metrics could be provided. We acknowledge that varying dataset sizes and different GPU architectures can make direct comparisons challenging due to discrepancies in training time and resource consumption; Our initial focus was on parameter counts (Table 4(b)) as a very intuitive measure because they remain fixed across various datasets and GPU architectures. However, to address these concerns, we have conducted additional experiments under consistent conditions to measure FLOPs, FPS, and training time per epoch. These metrics are provided below, along with comparisons to previous methods, to support our efficiency claims:

| Method           | Params   | % CLIP | Base  | Novel | HM    | FPS (batch 4) | GFLOPs | Time (1 epoch) |
|-------------------|----------|-----------------|-------|-------|-------|----------------|--------|------------------------|
| CoOp             | 2048     | 0.002 | 82.69 | 63.22 | 71.66 | 104.5| 162.5  | ~32s |
| CoCoOp           | 35360    | 0.03 | 80.47 | 71.69 | 75.83 |  53.3   | 162.5  | ~47s|
| MaPLe            | 3.55 M   | 2.85| 82.28 | 75.14 | 78.55 | 175.58| 167    | ~28s|
| ALIGN            | 3.58 M   | 2.87| 83.38 | 75.51 | 79.25 | 72.6| 314.6  | ~42s |
| PrompSRC         | 31488    | 0.02 | 84.26 | 76.10 | 79.97 | 149.86| 281.21 | ~27s |
| **DPD-LoRA†**    | **1.92 M** | **1.54** | **84.80** | **76.80** | **80.60** | **82.51**   | **334.03** | **~40s**|
| DPD-LoRA         | 4.72 M   | 3.79| 85.67 | 76.91 | 81.05 | 81.57| 334    | ~42s|

One more thing we hope reviewers may note is that even though our method has slightly higher GFLOPs due to additinal LoRA/LoRSS computations, **our convergence speed is much faster than any previous methods. Our method showcases accelerated convergence and favorable early-stage performance. Specifically, our method reaches better performance in just 7 epochs, which is 65% fewer epochs than the 20 epochs required by previous SOTA—a reduction of over 65% in training time (as shown in Figure 1b and Figure 5).**

---

### Author Response · Authors · 2024-11-17
**General Response to the Reviewers**

We sincerely thank the reviewers for their thoughtful and constructive feedback. We are encouraged by the positive recognition of our contribution, which can be summarized as follows:

1. DPD-LoRA shows **innovative exploration** in this domain (Reviewers `Tgia`, `zPfS`)
2. DPD-LoRA achieves **outstanding performance** (Reviewers `Tgia`, `pw2m`, `zPfS`)
3. Extensive experiments provide **convincing evidence of the effectiveness** (Reviewers `Tgia`, `pw2m`, `zPfS`)
4. The paper is **well-structured** and **easy to understand** (Reviewers `pw2m`, `1iPg`, `zPfS`)

In our revision, we have carefully addressed each of the concerns raised. Below are the main points of the revised manuscript(colored in blue text):

1. Corrected minor **typos** and **switched the introduction order** in the 'HIERARCHICAL INTERACTION AND EXPANDED SUBSPACES' sections; **merged more details** from the appendix into the section 'Prompt Learning with LoRA in Transformers'(Reviewers `1iPg`, `zPfS`).
2. **Reformatted our Equations** 6 and 7, as well as Table 4(a) in the appendix, to **unify/simplify hyper-parameter representation** and demonstrate that layer weights are the same (Reviewers `Tgia`, `pw2m`, `zPfS`).
3. Provided a **more concise introduction** to focus on our main contributions (i.e., prompt-guided adaptation and strengthening their connection with gating) (Reviewers `pw2m`, `zPfS`).
4. **Reformatted our Table** 5 to include more straightforward comparisons showing that LoRSS and our methods are better than previous methods and plain LoRA (Reviewers `Tgia`, `pw2m`); **reformatted our Table 4(b)** to include FPS in evaluation metrics (Reviewer `1iPg`).
5. Added **more references** to related work to reflect the progress in this area of LoRA (Reviewer `1iPg`).

Once again, **as the first** to explore how prompt learning can provide additional task-specific guidance to LoRA, we highly value the reviewers' insightful feedback and welcome any additional suggestions that can help us improve our work.

---

### Comment · Area_Chair_bKDT · 2024-11-22
**Interactive Discussions**

Dear Reviewers,

Thank you for your efforts in reviewing this paper. We highly encourage you to participate in interactive discussions with the authors before November 26, fostering a more dynamic exchange of ideas rather than a one-sided rebuttal.

Please feel free to share your thoughts and engage with the authors at your earliest convenience.

Thank you for your collaboration.

Best regards,
ICLR 2025 Area Chair

---

### Note · Authors · 2025-01-22

I have read and agree with the venue's withdrawal policy on behalf of myself and my co-authors.